# UNDERSTANDING AND MITIGATING EXTRAPOLATION FAILURES IN PHYSICS-INFORMED NEURAL NETWORKS

## ABSTRACT

Physics-informed Neural Networks (PINNs) have recently gained popularity due to their effective approximation of partial differential equations (PDEs) using deep neural networks (DNNs). However, their out of domain behavior is not well understood, with previous work speculating that the presence of high frequency components in the solution function might be to blame for poor extrapolation performance. In this paper, we study the extrapolation behavior of PINNs on a representative set of PDEs of different types, including high-dimensional PDEs. We find that failure to extrapolate is not caused by high frequencies in the solution function, but rather by shifts in the support of the Fourier spectrum over time. We term these *spectral shifts* and quantify them by introducing a Weighted Wasserstein-Fourier distance (WWF). We show that the WWF can be used to predict PINN extrapolation performance, and that in the absence of significant spectral shifts, PINN predictions stay close to the true solution even in extrapolation. Finally, we propose a transfer learning-based strategy to mitigate the effects of larger spectral shifts, which decreases extrapolation errors by up to $82\%$.

## 1 INTRODUCTION

Understanding the dynamics of complex physical processes is crucial in many applications in science and engineering. Oftentimes, these dynamics are modeled as partial differential equations (PDEs) that depend on time. In the PDE setting, we want to find a solution function $u(x, t)$ that satisfies a given governing equation of the form

$$f(x, t) := u_t + \mathcal{N}(u) = 0, x \in \Omega, t \in [0, T] \tag{1}$$

where $u_t := \frac{\partial u}{\partial t}$ denotes the partial derivative of $u$ with respect to time, $\mathcal{N}$ is a - generally nonlinear - differential operator, $\Omega \subset \mathbb{R}^d$, with $d \in \{1, 2, 3\}$ is a spatial domain, and $T$ is the final time for which we're interested in the solution. Moreover, we impose an initial condition $u(x, 0) = u^0(x)$, $\forall x \in \Omega$ on $u(x, t)$, as well as a set of boundary conditions. Together, these conditions specify the behaviors of the solution on the boundaries of the spatio-temporal domain.

Following the recent progress in deep learning, physics-informed neural networks (PINN) as introduced in Raissi et al. (2019) have garnered attention because of their simple, but effective way of approximating time-dependent PDEs with deep neural networks. PINNs preserve important physical properties described by the governing equations by parameterizing the solution and the governing equation simultaneously with a set of shared network parameters. After the success of the seminal paper Raissi et al. (2019), many sequels have applied PINNs to solve various PDE applications, e.g. Anitescu et al. (2019); Yang et al. (2021); Zhang et al. (2018); Doan et al. (2019). Physics-informed loss terms have also proven useful in machine learning more generally Davini et al.; Cai et al. (2021).

**Related work.** Most previous studies using the standard PINNs introduced in Raissi et al. (2019) have demonstrated the performances of their methods in interpolation only, i.e. on a set of testing points sampled within the same temporal range that the network was trained on. We refer to points sampled beyond the final time of the training domain as extrapolation. In principle, standard PINNs are expected to be able to learn the dynamics in Eq. (1) and, consequently, to approximate $u(x, t)$ accurately in extrapolation. However, previous work in Kim et al. (2020) and Bonfanti et al. (2023)

has shown that this is not the case: PINNs can deviate significantly from the true solution once they are evaluated in an extrapolation setting, calling into question their capability as a tool for learning the dynamics of physical processes.

From a foundational standpoint, studying extrapolation can therefore give us insights into the limitations of PINNs more generally. From a practical standpoint, constantly retraining PINNs from scratch when faced with a point that is outside their initial training domain is undesirable (Bonfanti et al. (2023); Zhu et al. (2022)), so anticipating whether their predictions remain accurate is crucial. Several recent papers have recognized the importance of the extrapolation problem in PINNs (Kapoor et al. (2023); Bonfanti et al. (2023); Cuomo et al. (2022); Kim et al. (2020)), and at least two have proposed methods to address it (Kim et al. (2020); Kapoor et al. (2023)). However, even a basic characterization of extrapolation behavior for PINNs trained to solve time-dependent PDEs is still absent from the literature. Previous works consider standard PINNs incapable of extrapolating beyond the training domain and suspect implicit biases in deep neural networks to lead to the learned solution becoming smooth or flat in extrapolation, thus implying that the presence of high frequencies in the solution function might lead to extrapolation failures (Bonfanti et al. (2023)). Finally, there are to the best of our knowledge no theoretical works on the extrapolation capabilities of PINNs. Previous works have focused on PINN generalization in interpolation only (Mishra and Molinaro (2022)).

**Contributions.** In this paper, our contributions are therefore as follows. (i) We show that PINNs are capable of almost perfect extrapolation behavior for certain PDEs. (ii) We characterize these PDEs by analyzing the Fourier spectra of their solution functions and argue that standard PINNs generally fail to anticipate shifts in the support of the Fourier spectrum over time. We quantify these spectral shifts using the Wasserstein-Fourier distance. (iii) We clarify that unlike with training failures in interpolation, the presence of high frequencies alone is not to blame for the poor extrapolation behavior of PINNs on some PDEs. (iv) We show that these insights generalize to high-dimensional PDEs, and (v) we demonstrate the transfer learning on a set of similar PDEs can reduce extrapolation errors significantly when spectral shifts are present.

The structure of the paper is as follows: in section 2, we formally introduce PINNs and define what we mean by interpolation and extrapolation. Section 3 characterizes the PDEs for which good extrapolation accuracy is possible using the Fourier spectra of their solution functions and introduce the Weighted Wasserstein-Fourier distance. In section 4, we investigate the viability of transfer learning approaches in improving extrapolation. Section 5 discusses our results and concludes.

## 2 BACKGROUND AND DEFINITIONS

**Physics-Informed Neural Networks.** As mentioned in the previous section, PINNs parameterize both the solution $u$ and the governing equation $f$. Denote the neural network approximating the solution $u(x, t)$ by $\tilde{u}(x, t; \theta)$ and let $\theta$ be the network's weights. Then the governing equation $f$ is approximated by a neural network $\tilde{f}(x, t, \tilde{u}; \theta) := \tilde{u}_t + \mathcal{N}(\tilde{u}(x, t; \theta))$. The partial derivatives here can be obtained via automatic differentiation. We note that $\tilde{f}(x, t, \tilde{u}; \theta)$ shares its network weights with $\tilde{u}(x, t; \theta)$. The name "physics-informed" neural network comes from the fact that the physical laws we're interested in are enforced by applying an extra, problem-specific, nonlinear activation, which is defined by the PDE in Eq. (1) (i.e., $\tilde{u}_t + \mathcal{N}(\tilde{u})$).

We learn the shared network weights using a loss function consisting of two terms, which are associated with approximation errors in $\tilde{u}$ and $\tilde{f}$, respectively. Raissi et al. (2019) considers a loss of the form $L := \alpha L_u + \beta L_f$, where $\alpha, \beta \in \mathbb{R}$ are coefficients and $L_u$ and $L_f$ are defined as follows:

$$L_u = \frac{1}{N_u} \sum_{i=1}^{N_u} \left| u(x_u^i, t_u^i) - \tilde{u}(x_u^i, t_u^i; \theta) \right|; \; L_f = \frac{1}{N_f} \sum_{i=1}^{N_f} \left| \tilde{f}(x_f^i, t_f^i, \tilde{u}; \theta) \right|^2 \qquad (2)$$

$L_u$ enforces the initial and boundary conditions using a set of training data $\left\{ (x_u^i, t_u^i), u(x_u^i, t_u^i) \right\}_{i=1}^{N_u}$. The first element of the tuple is the input to the neural network $\tilde{u}$ and the second element is the ground truth that the output of $\tilde{u}$ attempts to match. We can collect this data from the specified initial and boundary conditions since we know them a priori. Meanwhile, $L_f$ minimizes the discrepancy between the governing equation $f$ and the neural network's approximation $\tilde{f}$. We evaluate the network

at collocation points $\left\{(x_f^i, t_f^i), f(x_f^i, t_f^i)\right\}_{i=1}^{N_f}$. Note that here, the ground truth $\left\{f(x_u^i, t_u^i)\right\}_{i=1}^{N_f}$ consists of all zeros. We also refer to $\frac{1}{N_f} \sum_{i=1}^{N_f} \left|\tilde{f}(x_f^i, t_f^i, \tilde{u}; \theta)\right|$ as the mean absolute residual (MAR): its value denotes how far the network is away from satisfying the governing equation. Note that using this loss, i) no costly evaluations of the solutions $u(x, t)$ at collocation points are required to gather training data, ii) initial and boundary conditions are enforced using a training dataset that can easily be generated, and iii) the physical law encoded in the governing equation $f$ in Eq. (1) is enforced by minimizing $L_f$. In the original paper by Raissi et al. (2019), both loss terms have equal weight, i.e. $\alpha = \beta = 1$, and the combined loss term $L$ is minimized.

**Interpolation and extrapolation.** For the rest of this paper, we refer to points $(x^i, t^i)$ as interpolation points if $t^i \in [0, T_{\text{train}}]$, and as extrapolation points if $t^i \in (T_{\text{train}}, T_{\text{max}}]$ for $T_{\text{max}} > T_{\text{train}}$. We are primarily interested in the $L^2$ error of the learned solution, i.e. in $\|u(x^i, t^i) - \tilde{u}(x^i, t^i; \theta)\|_2$, and in the $L^2$ relative error, which is the $L^2$ error divided by the norm of the function value at that point, i.e. $\|u(x^i, t^i)\|_2$. When we sample evaluation points from the extrapolation domain, we refer to the $L^2$ (relative) error as the (relative) extrapolation error. Similarly, we are interested in the (mean) absolute residual as defined above, i.e. in $\left|\tilde{f}(x^i, t^i, \tilde{u}; \theta)\right|$. For points sampled from the extrapolation domain, we refer to this as the extrapolation residual.

In this paper, we are interested in the extrapolation performance of PINNs, by which we broadly mean the following questions: how quickly does the performance of a PINN deteriorate as we move away from the interpolation domain? What aspects of the model or underlying PDE affect this? When we speak of "near perfect" extrapolation, we therefore always mean the accuracy of the model on a bounded extrapolation domain, usually neighboring the interpolation domain. This is in line with Kim et al. (2020) and distinct from the question whether MLPs more generally can extrapolate to arbitrary domains Haley and Soloway (1992); Cardell et al. (1994); Ziyin et al. (2020).

**PDEs considered.** We investigate the extrapolation capabilities of PINNs on a representative set of 7 PDEs, all of which are widely used as examples in the PINN literature Basir (2022); Raissi et al. (2019); Penwarden et al. (2023); Jagtap and Karniadakis (2021). These include the Allen-Cahn equation, the viscous Burger's equation, a heat equation, a diffusion equation, a diffusion-reaction equation, the Beltrami flow, and the non-linear Schrodinger equation. Details on all PDEs considered can be found in Appendix A.1.

## 3 UNDERSTANDING EXTRAPOLATION FAILURES VIA SPECTRAL SHIFTS

### 3.1 EFFECTS OF MODEL SIZE, ACTIVATION FUNCTIONS, & NUMBER OF TRAINING SAMPLES

Before we begin our investigation of what determines extrapolation performance in PINNs, we identify several aspects of a model that **do not** have an effect. This will make our analysis in the second half of this section easier. To this end, we analyze the extrapolation errors and residuals which standard PINNs display for the Allen-Cahn equation, the viscous Burgers equation, a diffusion equation, and a diffusion-reaction equation.

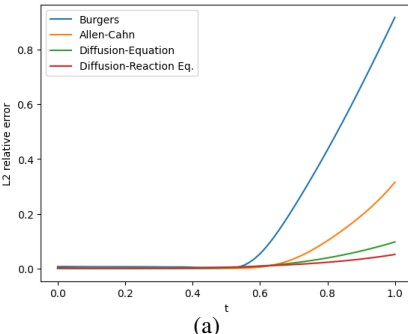
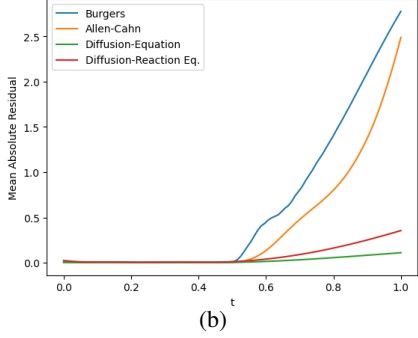

(a)  (b)

Figure 1: **(a)** $L^2$ relative extrapolation error of MLP(5, 64) with $tanh$ activation, trained on $[0, 0.5]$. and **(b)** MAR for the same MLP.

**PINN extrapolation performance depends on the underlying PDE.** For each of the four PDEs introduced above, we train a 5-layer MLP with 64 neurons per layer and $\tanh$ activation on the interpolation domains specified for 50000 epochs using the adam optimizer. As seen in Figure 1 observe that the $L^2$ relative errors for the Burgers' equation and for the Allen-Cahn equation become significantly larger than for the diffusion and diffusion-reaction equations when we move from $t = 0.5$ to $t = 1$. The solution learned for the diffusion-reaction equation disagrees only minimally with the true solution, even at $t = 1$, which shows that for this particular PDE, PINNs can extrapolate almost perfectly well. More detailed results can be found in Appendix A.2.

**Extrapolation performance is generally independent of model parameters.** While we observe drastically different extrapolation behaviors depending on the underlying PDE as mentioned above, the extrapolation for a given PDE seems to be more or less independent of model parameters, such as number of layers or neurons per layer, activation function, number of samples, or training time. Once the chosen parameters allow the model to achieve a low error in the interpolation domain - $1e - 5$ is a value commonly used for this in the literature Raissi et al. (2019); Chen et al. (2023); Wang et al. (2022); Han and Lee (2021) - adding more layers, neurons, or samples, or alternatively training longer does not seem to have an effect on the extrapolation error and MAR.

These results allow us to focus our further analyses on a single architecture. Unless otherwise stated, we use an MLP with 5 layers with 64 neurons each and $\tanh$ activation, initialized with the commonly used Xavier normal initialization, and trained for 50000 epochs using Adam.

### 3.2 EXTRAPOLATION IN THE PRESENCE OF HIGH FREQUENCIES

Recent literature has found that neural networks tend to be biased towards low-complexity solutions due to implicit regularization inherent in their gradient descent learning processes Neyshabur et al. (2014); Neyshabur (2017). In particular, deep neural networks have been found to possess an inductive bias towards learning lower frequency functions, a phenomenon termed the *spectral bias* of neural networks Rahaman et al. (2019); Cao et al. (2019), which for example Bonfanti et al. (2023) suspect to be related to extrapolation failures in PINNs. They find evidence for this when considering time-independent PDEs.

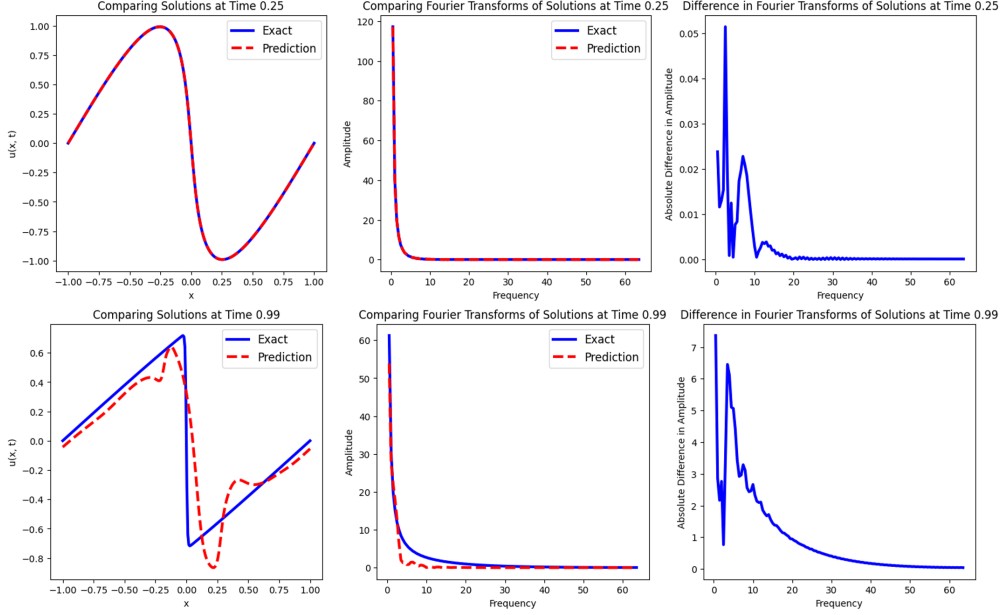

Figure 2: For times $t = 0.25$ (top, interpolation) and $t = 0.99$ (bottom, extrapolation), we plot the reference and predicted solutions in the spatio-temporal (left) and Fourier (middle) domains for the Burgers' equation. The absolute difference in the Fourier spectra is plotted on the right.

Following this hypothesis, we would expect most of the extrapolation error to come from the higher frequencies: the predicted function might become smooth or flat in extrapolation, similar to what

has been observed with training failures in interpolation Basir (2022). We plot both the reference solution and the predicted solution in the Fourier domain for all four of our PDEs, as well as the absolute difference between the two Fourier spectra of the reference and predicted solution. Plots for the Burgers' equation are provided in Figure 2 while plots for the other PDEs are provided in Appendix A.3.

**High frequencies only account for a small fraction of extrapolation errors.** In all cases, the majority of the error in the Fourier domain is concentrated in the lower-frequency regions. While this is partially due to the fact that the low frequency components of the solutions have larger magnitude, it suggests that in extrapolation, PINNs fail even to learn the low frequency parts of the solution. Thus, the presence of high frequencies alone fails to explain the extrapolation failure of PINNs. We provide some additional evidence for this by studying the extrapolation behavior of Multi-scale Fourier feature networks Wang et al. (2020) in Appendix A.6. Even though these architectures were designed specifically to make learning higher frequencies easier, we find their extrapolation error to be at least as large or larger than that of standard PINNs.

**PINNs can extrapolate well in the presence of high frequencies.** To isolate the effect that the presence of high frequencies **alone** has on extrapolation performance, we consider the following variation of the Diffusion-Reaction for $x \in [-\pi, \pi]$ and $t \in [0, 1]$.

$$\frac{\partial u}{\partial t} = \frac{\partial^2 u}{\partial x^2} + e^{-t} \left( \sum_{j=1}^{K} \frac{(j^2 - 1)}{j} \sin(jx) \right) \tag{3}$$

$$u(x, 0) = \sum_{j=1}^{K} \frac{\sin(jx)}{j} \qquad u(-\pi, t) = u(\pi, t) = 0 \tag{4}$$

The reference solution is given by $u(x, t) = e^{-t} \left( \sum_{j=1}^{K} \frac{\sin(jx)}{j} \right)$. As with our other experiments, we use $t \in [0, 0.5]$ as the temporal training domain and consider $t \in (0.5, 1]$ as the extrapolation area. $K$ here is a hyperparameter that controls the size of the spectrum of the solution. Note that for a fixed $K$, the support of the Fourier spectrum of the reference solution never changes over time, with only the amplitudes of each component scaled down by an identical constant factor.

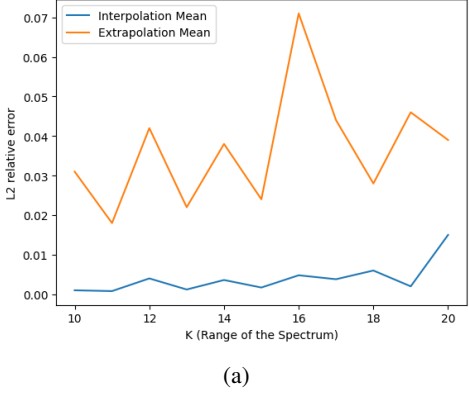

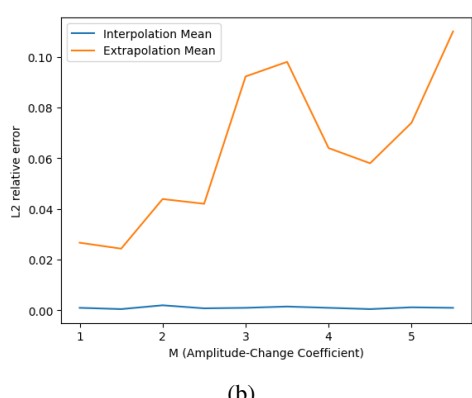

| (a) | (b) |

Figure 3: Mean $L^2$ relative interpolation and extrapolation errors, trained on $[0, 0.5]$. In **(a)**, we plot this against the size of the spectrum i.e. the parameter $K$ in Equation (4), and in **(b)** we plot this against the speed of the decay of the amplitudes, i.e. the parameter $M$ in Equation (6).

For various values of $K$, we find that our trained PINNs are able to extrapolate well as can be seen in Figure 3 **(a)**. For the sake of completeness, we also investigate the effect of the speed of decay of the amplitudes in the Fourier spectra. We train a PINN on the following variation of the Diffusion-Reaction equation.

$$\frac{\partial u}{\partial t} = \frac{\partial^2 u}{\partial x^2} + e^{-Mt} \left( \sum_{j \in \{1,2,3,4,8\}} \frac{(j^2 - 1)}{j} \sin(jx) \right) \tag{5}$$

for $x \in [-\pi, \pi]$ and $t \in [0, 1]$ with the initial condition $u(x, 0) = \sin(x) + \frac{\sin(2x)}{2} + \frac{\sin(3x)}{3} + \frac{\sin(4x)}{4} + \frac{\sin(8x)}{8}$ and the Dirichlet boundary condition $u(-\pi, t) = u(\pi, t) = 0$. The reference solution is

$$u(x, t) = e^{-Mt} \left( \sin(x) + \frac{\sin(2x)}{2} + \frac{\sin(3x)}{3} + \frac{\sin(4x)}{4} + \frac{\sin(8x)}{8} \right) \tag{6}$$

with the same interpolation and extrapolation areas as before. Figure 3 **(b)** shows the relative interpolation and extrapolation errors against increasing values of $M$. We find that an increase in the speed of the exponential decay seems to increase the extrapolation error more than an increase in the size of the spectrum.

### 3.3 SPECTRAL SHIFTS

While the solutions to the Allen-Cahn equation and to the Burger's equation do not exhibit exponentially fast changes in their amplitudes, they have Fourier spectra whose support shifts over time, unlike the diffusion and diffusion-reaction equations. We argue that PINNs struggle to extrapolate well when these *spectral shifts* in the true solution's Fourier spectrum are large.

**Weighted Wasserstein-Fourier distance.** To quantify the temporal shifts in the support of the Fourier spectrum, we introduce the *Weighted Wasserstein-Fourier Distance* (WWF) between the normalized Fourier spectra of the PDE solution in two disjoint time domains. The Weighted Wasserstein-Fourier Distance is based on the Wasserstein-Fourier distance, which compares the Fourier spectra of the solution function at two different points in time. Consider two discrete CDFs $F_1, F_2$ supported on the domain $\mathcal{X}$. The Wasserstein distance between $F_1$ and $F_2$ is defined as $W(F_1, F_2) = \sum_{x \in \mathcal{X}} |F_1(x) - F_2(x)|$. Given two discrete Fourier spectra $f_1, f_2$, the Wasserstein-Fourier distance Cazelles et al. (2020) can be computed as $W\left( \frac{f_1}{\|f_1\|_1}, \frac{f_2}{\|f_2\|_1} \right)$. We now define the Weighted Wasserstein-Fourier Distance given a function $f$ as

$$WWF(f) := \sum_{s \in I} \sum_{t \in E} (T_{max} + s - t) W \left( \frac{f_s}{\|f_s\|_1}, \frac{f_t}{\|f_t\|_1} \right)$$

where $I$ and $E$ are the interpolation and extrapolation domains, respectively. We present plots of the pairwise Wasserstein-Fourier distances for each $t_1, t_2 \in [0, T_{max}]$ in Appendix A.4. The Wasserstein-Fourier distance of the true solution is zero everywhere for both the diffusion and diffusion-reaction equations, leading to a Weighted Wasserstein-Fourier Distance of zero, which reflects the constant support of the spectra. In contrast, the pairwise distance matrices for the Burgers' and Allen-Cahn equations exhibit a block-like structure, with times in disjoint blocks exhibiting pronouncedly different distributions in the amplitudes of their respective Fourier spectra. These shifts are not captured by the learned solution, leading to large $L^2$ errors.

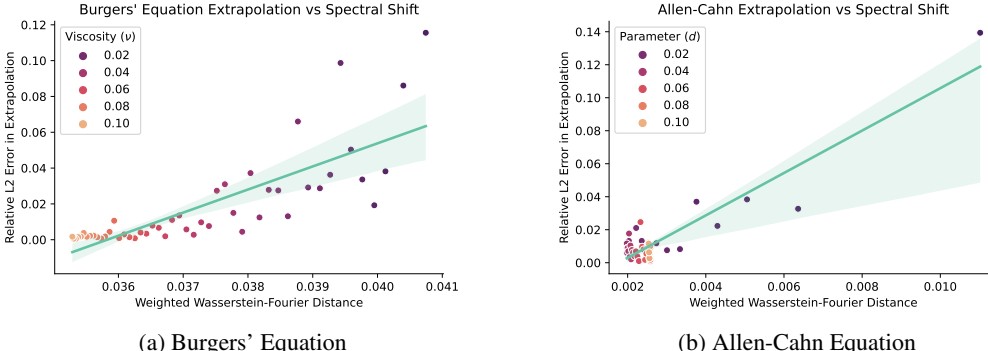

(a) Burgers' Equation                    (b) Allen-Cahn Equation

Figure 4: For both Burgers' Equation (a) and the Allen-Cahn equation (b), we train 50 PINNs on a variety of different PDE parameters for each equation. More extreme spectral shifts in the underlying solution are correlated with poorer extrapolation performance.

The Weighted Wasserstein-Fourier distance allows us to capture the effects that other properties of the underlying PDE have on extrapolation performance. To illustrate this, we train PINNs for 50 different Burgers' equations, each with a different viscosity parameter $\nu$ – equally spaced from 0.001 to 0.1, and for 50 variants of the Allen-Cahn equation with varying values of $d$, equally spaced from 0.0001 to 0.1. We find that different PDE coefficients lead to large differences in extrapolation performance and that this relationship is moderated quite heavily through shifts in the underlying Fourier spectra. Figure 4 plots the WWF distance between the spectra against the relative $L^2$ error in extrapolation. PDE coefficients that induce larger shifts in the spectra correspond to overall worse extrapolation performance.

### 3.4 Higher-dimensional and more complex PDEs

Our findings so far demonstrate that the extrapolation performance of PINNs depends heavily on the presence of spectral shifts in the underlying PDE. We conclude this section by showing that this remains true for higher-dimensional and more complex PDEs. To this end, we train PINNs on the Beltrami Flow and the non-linear Schrodinger equation.

The reference solution to the non-linear Schrodinger equation exhibits significant shifts in the spectra, with a WWF Distance distance between the interpolation and extrapolation domains of 0.034 and 0.036 in the real and imaginary domain respectively. Based on our results for lower-dimensional PDEs, we expect extrapolation performance to be poor. Our experimental results agree: while the PINN achieves a small interpolation error ($1e-5$), it exhibits poor extrapolation behavior, achieving max $L^2$ relative errors of 0.94, and 4.27 (in the real and imaginary domain respectively).

On the other hand, the Beltrami flow does not exhibit a spectral shift over time for any of the solution functions. The PINN achieves similarly small interpolation error ($1e-5$) and produces very small $L^2$ relative extrapolation errors of 0.009, 0.013, 0.006, and 0.008 in $u, v, w$, and $p$, respectively. This is in line with what we would expect based on the lower-dimensional examples consider so far, and is in fact comparable to the diffusion-reaction equation.

## 4 Mitigating extrapolation failures with transfer learning

Finally, we show that transfer learning from PINNs trained across a family of similar PDEs can improve extrapolation performance. Empirically, in other domains, transfer learning across multiple tasks has been effective in improving generalization Dong et al. (2015); Luong et al. (2016). Here, we perform transfer learning following the procedure outlined in Pellegrin et al. (2022), where we initially train a PINN with multiple outputs on a sample from a family of PDEs (e.g. the Burgers' equation with varying values of the viscosity) and transfer to a new unseen PDE in the same family (e.g. the Burgers' equation with a different viscosity) by freezing all but the last layer and training with the loss this new PDE induces. We note that Pellegrin et al. (2022) only consider transfer learning for linear PDEs by analytically computing the final PINN layer but we extend their method to nonlinear PDEs by performing gradient descent to learn the final layer instead.

### 4.1 Transfer learning can help with spectral shifts

We perform transfer learning from a collection of Burgers' equations with varying viscosities ($\nu/\pi = \{0.01, 0.05, 0.1\}$) to a new Burgers equation ($\nu/\pi = 0.075$). In the first set of experiments, we train on equations in the domain $t \in [0, 0.5]$, and in the second set, we train on equations in the domain $t \in [0, 1]$. Similarly, for the non-linear Schrodinger equation we transfer learn on equations with slightly varying initial conditions ($h(x, 0) \in \{1.95\text{sech}(x), 2.05\text{sech}(x), 2.1\text{sech}(x)\}$). We evaluate on a new non-linear Schrodinger equation with initial condition $h(x, 0) = 2\text{sech}(x)$. Our results are reported in Table 1. We perform 15 runs for each, changing only the random seed.

Compared to the baseline (no transfer learning), we find an average reduction in extrapolation error of $82\%$ when transfer learning from the full domain, and of $51\%$ when transfer learning from half the domain, i.e. with $t \in [0, 0.5]$ for the Burger's equation. The improvements for the non-linear Schrodinger equation are similar, although slightly smaller. transfer learning from the full domain reduces the extrapolation error in the real (imaginary) component of the solution by $55\%$ ($51\%$). Transfer learning from half the domain still reduces it by $32\%$ ($30\%$). Details on the same transfer

| | L² Relative Extrapolation Error | | |
|---|---|---|---|
| Setting | Burger's Eq. | Schrodinger (real) | Schrodinger (imag.) |
| Baseline | $0.383 \pm 0.143$ | $0.944 \pm 0.212$ | $4.276 \pm 0.538$ |
| Transfer (half) | $0.189 \pm 0.116$ | $0.630 \pm 0.227$ | $2.963 \pm 0.599$ |
| Transfer (full) | $0.072 \pm 0.065$ | $0.423 \pm 0.201$ | $2.074 \pm 0.526$ |

Table 1: $L^2$ extrapolation errors for the baseline (no transfer learning), transfer learning from $t \in [0, 0.5]$ (half), and transfer learning from $t \in [0, 1]$ (full). Values obtained from 15 MLPs per setting.

learning experiments for the Allen-Cahn equation, as well as visualizations of the learned solutions can be found in subsection A.8 in the appendix.

**Why does transfer learning help?** By transfer learning from other PDEs that exhibit similar spectral shifts, we hope that the model can learn to recognize PDEs that exhibit these shifting spectra and modify its predictions accordingly. As we freeze all but the last layer when performing transfer learning, one can think of this as projecting the new PDE onto a shared feature space, one of these features potentially capturing the degree to which the underlying spectra shift over time. Given that the initial training is conducted on a larger temporal domain, the hope is that even if the model is trained on a new PDE only from $t = 0$ to $t = 0.5$, its understanding of frequency shifts from similar PDEs (for which it knows how the spectra evolve/shift from $t = 0$ to $t = 1$) will allow it to extrapolate better than it otherwise would.

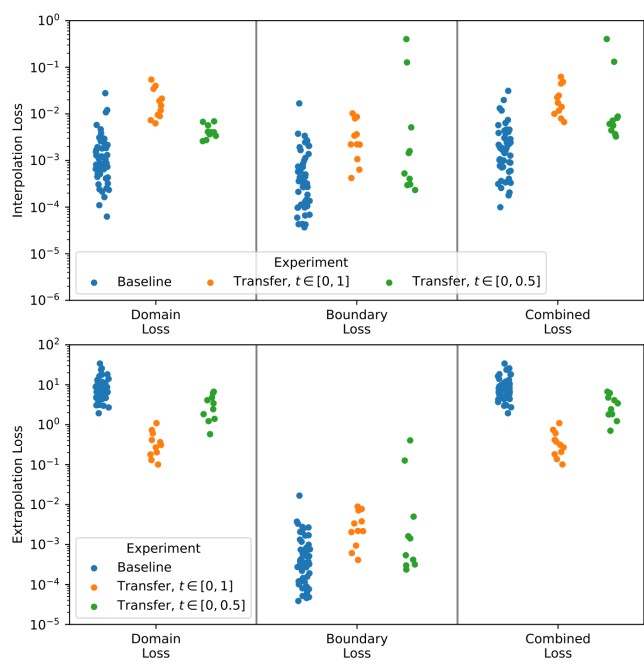

Figure 5: Domain, boundary, and combined mean squared interpolation error (top) and extrapolation error (bottom) between our baseline (PINNs trained from scratch) and transfer learning experiments. The only variation between data points is the random seed. There are 50, 11, and 10 runs of the baseline, transfer with $t \in [0, 1]$, and transfer with $t \in [0, 0.5]$, respectively. Note the vertical scales differ between the interpolation and extrapolation domains.

To give some evidence to support this intuition, our transfer learning experiments use Burgers' equations with similar viscosities ($\nu$) to the target PDE – and thus similar spectral shift. We find that additional transfer learning on more PDEs, with viscosities that are further from that of the target PDE, seems to make a minimal impact.

Motivated by Kim et al. (2020), we can also examine the interpolation and extrapolation loss of each run as well as decomposed into domain and boundary terms (recall Section 2) using the example of the Burger's equation in Figure 5. We observe that transfer learning from PDEs on the whole domain ($t \in [0, 1]$) substantially improves results compared to baseline. However, we find that transfer learning even when the model does not see the extrapolation domain during initial training (e.g. $t \in [0, 0.5]$) also improves performance over baseline, though less than transfer learning from the full domain. We find the reverse in interpolation: our baseline model has the lowest interpolation error, followed by half-domain transfer learning, and then full-domain transfer learning, which performs

the worst in interpolation. This may suggest that transfer learning enforces stronger inductive biases from the wider PDE family which in turn improves extrapolation performance.

## 4.2 WITHOUT SPECTRAL SHIFTS, TRANSFER LEARNING YIELDS NO IMPROVEMENTS

We repeat the experiments in the previous subsection with PDEs that exhibit no spectral shifts to test whether transfer learning can further boost extrapolation performance. We transfer learn on Diffusion-Reaction equations with different amplitude parameters (recall section 4, here $M = \{0.5, 2, 3\}$) and evaluate on a Diffusion-Reaction equation with amplitude parameter $M = 1$. As a high-dimensional analogue, we transfer learn on the Beltrami Flow PDE with $Re = \{0.95, 1.05, 1.1\}$ and evaluate on $Re = 1$. We present our results in table 2.

| | $\mathbf{L^2}$ **Relative Extrapolation Error** | | | | |
|---|---|---|---|---|---|
| **Setting** | **Diff.-Reac.** | **Beltrami (u)** | **Beltrami (v)** | **Beltrami (w)** | **Beltrami (p)** |
| Baseline | $0.038 \pm 0.021$ | $0.009 \pm 0.004$ | $0.013 \pm 0.006$ | $0.006 \pm 0.003$ | $0.008 \pm 0.004$ |
| Transfer (half) | $0.051 \pm 0.033$ | $0.011 \pm 0.005$ | $0.009 \pm 0.007$ | $0.007 \pm 0.003$ | $0.006 \pm 0.003$ |
| Transfer (full) | $0.043 \pm 0.024$ | $0.008 \pm 0.004$ | $0.012 \pm 0.007$ | $0.006 \pm 0.005$ | $0.007 \pm 0.005$ |

Table 2: $L^2$ extrapolation errors for the baseline (no transfer learning), transfer learning from $t \in [0, 0.5]$ (half), and transfer learning from $t \in [0, 1]$ (full). Values obtained from 15 MLPs per setting.

Unlike with the PDEs in the previous section, which showed a significant spectral shift, we find no improvement in extrapolation performance for the Diffusion-Reaction equation or the Beltrami Flow after transfer learning. In line with our reasoning for why transfer learning helps with spectral shifts, we suspect that because there are no spectral shifts in any of the PDEs considered, there is nothing for the model to pick up while transfer learning. Similarly, in the absence of spectral shifts, stronger inductive biases need not improve extrapolation, and in fact might make it harder.

## 5 DISCUSSION

In this paper, we revisited PINNs' extrapolation behavior and pushed back against claims previously made in the literature. In our experiments on the effects of different architecture choices, we found evidence against a double-descent phenomenon for the extrapolation error, which Zhu et al. (2022) speculated might exist. We also saw that PINNs do not necessarily perform poorly in extrapolation, as was previously suspected (Kim et al. (2020); Kapoor et al. (2023)). For some PDEs, near perfect extrapolation is possible. Following this, we examined the solution space learned by PINNs in the Fourier domain and argued that extrapolation performance depends on spectral shifts in the underlying PDE. We showed that the presence of high frequencies in the solution function has minimal effect on extrapolation, pushing back against Bonfanti et al. (2023), and demonstrated that PINNs' extrapolation errors can be predicted from the Fourier spectra of the solution function. To this end, we introduced the Weighted Wasserstein-Fourier distance between interpolation and extrapolation domains. Finally, we provided the first investigation of the effects of transfer learning on extrapolation behavior in PINNs and demonstrated that transfer learning can help mitigate the effects of spectral shifts.

**Limitations.** There are several avenues for further investigation. We believe that extending our analysis from standard PINNs to other architectures or sampling methods is a promising direction. Future research might, for example, try to answer whether some PINN variants can deal better with spectral shifts than others and why. Furthermore, in the present work, we only examined the two most common activation functions, $\sin$ and $\tanh$, and found them to lead to similar model performance in extrapolation. While this is in line with experiments presented in related works Kim et al. (2020), investigating activation functions specifically introduced for improved extrapolation performance in MLPs, such as Ziyin et al. (2020), could also prove insightful. Ultimately, we believe that a theoretical investigation of PINNs' difficulties with spectral shifts in the fashion of Wang et al. (2020) could significantly deepen our understanding of these models' capabilities.

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

# A    APPENDIX

CONTENTS

### A.1 PDEs under Consideration

#### A.1.1 Viscous Burger's Equation

The viscous Burger's equation is given by

$$\frac{\partial u}{\partial t} + u\frac{\partial u}{\partial x} = \nu\frac{\partial^2 u}{\partial x^2} \tag{7}$$

Here, we consider $x \in [-1, 1]$ and $t \in [0, 1]$. We set $\nu = 0.01$ and use the Dirichlet boundary conditions and initial conditions

$$u(-1, t) = u(1, t) = 0 \,,\, u(x, 0) = -\sin(\pi x) \tag{8}$$

We consider $t \in [0, 0.5]$ as the interpolation domain and $t \in (0.5, 1]$ as the extrapolation domain.

#### A.1.2 Allen-Cahn Equation

The Allen-Cahn equation is of the form

$$\frac{\partial u}{\partial t} = d \cdot \frac{\partial^2 u}{\partial x^2} + 5\left(u - u^3\right) \tag{9}$$

for $x \in [-1, 1]$ and $t \in [0, 1]$ We set $d = 0.001$ and consider $t \in [0, 0.5]$ as the interpolation domain and $(0.5, 1]$ as the extrapolation domain. The initial and the boundary conditions are given by

$$u(x, 0) = x^2\cos(\pi x); u(-1, t) = u(1, t) = -1 \tag{10}$$

#### A.1.3 Diffusion Equation

We consider the diffusion equation

$$\frac{\partial u}{\partial t} = \frac{\partial^2 u}{\partial x^2} - e^{-t}\left(\sin(\pi x) - \pi^2\sin(\pi x)\right) \tag{11}$$

for $x \in [-1, 1]$ and $t \in [0, 1]$ with the initial condition $u(x, 0) = \sin(\pi x)$ and the Dirichlet boundary condition $u(-1, t) = u(1, t) = 0$. The reference solution is $u(x, t) = e^{-t}\sin(\pi x)$. We use $t \in [0, 0.5]$ as the temporal training domain and consider $t \in (0.5, 1]$ as the extrapolation area.

#### A.1.4 Diffusion-Reaction Equation

The diffusion-reaction equation we consider is closely related to the diffusion equation above, but has a larger Fourier spectrum. Formally, we consider

$$\frac{\partial u}{\partial t} = \frac{\partial^2 u}{\partial x^2} + e^{-t}\left(3\frac{\sin(2x)}{2} + 8\frac{\sin(3x)}{3} + 15\frac{\sin(4x)}{4} + 63\frac{\sin(8x)}{8}\right) \tag{12}$$

for $x \in [-\pi, \pi]$ and $t \in [0, 1]$ with the initial condition

$$u(x, 0) = \sin(x) + \frac{\sin(2x)}{2} + \frac{\sin(3x)}{3} + \frac{\sin(4x)}{4} + \frac{\sin(8x)}{8} \tag{13}$$

and the Dirichlet boundary condition $u(-\pi, t) = u(\pi, t) = 0$. The reference solution is

$$u(x, t) = e^{-t}\left(\sin(x) + \frac{\sin(2x)}{2} + \frac{\sin(3x)}{3} + \frac{\sin(4x)}{4} + \frac{\sin(8x)}{8}\right) \tag{14}$$

We consider the same interpolation and extrapolation domains as before.

### A.1.5 HEAT EQUATION

The heat equation we consider is given by

$$\frac{\partial u}{\partial t} = \alpha \frac{\partial^2 u}{\partial x^2} \tag{15}$$

with thermal diffusivity coefficient $\alpha = 0.4$ and $x \in [-1,1], t \in [0,1]$. The Dirichlet boundary conditions are

$$u(0,t) = u(1,t) = 0 \tag{16}$$

and the initial condition is given by

$$u(x,0) = \sin(\pi x) \tag{17}$$

The exact solution is

$$u(x,t) = e^{\pi^2 \alpha t} \sin(\pi x) \tag{18}$$

### A.1.6 BELTRAMI FLOW

We consider the following Beltrami flow PDE:

$$\frac{\partial u}{\partial t} + \left( u\frac{\partial u}{\partial x} + v\frac{\partial u}{\partial y} + w\frac{\partial u}{\partial z} \right) + \frac{\partial p}{\partial x} - \frac{1}{Re}\left( \frac{\partial^2 u}{\partial x^2} + \frac{\partial^2 u}{\partial y^2} + \frac{\partial^2 u}{\partial z^2} \right) = 0$$

$$\frac{\partial v}{\partial t} + \left( u\frac{\partial v}{\partial x} + v\frac{\partial v}{\partial y} + w\frac{\partial v}{\partial z} \right) + \frac{\partial p}{\partial y} - \frac{1}{Re}\left( \frac{\partial^2 v}{\partial x^2} + \frac{\partial^2 v}{\partial y^2} + \frac{\partial^2 v}{\partial z^2} \right) = 0$$

$$\frac{\partial w}{\partial t} + \left( u\frac{\partial w}{\partial x} + v\frac{\partial w}{\partial y} + w\frac{\partial w}{\partial z} \right) + \frac{\partial p}{\partial z} - \frac{1}{Re}\left( \frac{\partial^2 w}{\partial x^2} + \frac{\partial^2 w}{\partial y^2} + \frac{\partial^2 w}{\partial z^2} \right) = 0$$

$$\frac{\partial u}{\partial x} + \frac{\partial v}{\partial y} + \frac{\partial w}{\partial z} = 0$$

for $(x,y,z) \in [-1,1]^3, t \in [0,1]$ and with Dirichlet boundary conditions. The solution functions are given by

$$u(x,y,z,t) = -a[e^{ax}\sin(ay+dz) + e^{az}\cos(ax+dy)]e^{-d^2 t} \tag{19}$$

$$v(x,y,z,t) = -a[e^{ay}\sin(az+dx) + e^{ax}\cos(ay+dz)]e^{-d^2 t} \tag{20}$$

$$w(x,y,z,t) = -a[e^{az}\sin(ax+dy) + e^{ay}\cos(az+dx)]e^{-d^2 t} \tag{21}$$

$$\begin{aligned}
p(x,y,z,t) = -0.5a^2[&e^{2ax} + e^{2ay} + e^{2az} + 2\sin(ax+dy)\cos(az+dx)e^{a(y+z)} \\
&+ 2\sin(ay+dz)\cos(ax+dy)e^{a(z+x)} \\
&+ 2\sin(az+dx)\cos(ay+dz)e^{a(x+y)}]e^{(-2d^2 t)}
\end{aligned} \tag{22}$$

where $a = d = Re = 1$ unless explicitly stated otherwise. We consider the same interpolation and extrapolation domains as before.

### A.1.7    Non-linear Schrodinger equation

The Nonlinear Schrodinger equation we consider is defined as

$$i\frac{\partial h}{\partial t} + \frac{1}{2}\frac{\partial^2 h}{\partial x^2} + |h|^2 h = 0 \tag{23}$$

subject to the periodic boundary conditions $x \in [-5, 5], h(t, -5) = h(t, 5), h_x(t, -5) = h_x(t, 5)$ and the initial condition $h(0, x) = 2\text{sech}(x)$. We use $t \in [0, \pi/4]$ as the interpolation domain and $t \in (\pi/4, \pi/2]$ as the extrapolation domain.

### A.1.8    Navier-Stokes equations in two Dimensions

The Navier-Stokes equations in two dimensions are given explicitly by

$$\frac{\partial u}{\partial t} + u\frac{\partial u}{\partial x} + v\frac{\partial u}{\partial y} + \frac{\partial p}{\partial x} - \nu\left(\frac{\partial^2 u}{\partial x^2} + \frac{\partial^2 u}{\partial y^2}\right) = 0$$

$$\frac{\partial v}{\partial t} + u\frac{\partial v}{\partial x} + v\frac{\partial v}{\partial y} + \frac{\partial p}{\partial y} - \nu\left(\frac{\partial^2 v}{\partial x^2} + \frac{\partial^2 v}{\partial y^2}\right) = 0$$

$$\frac{\partial u}{\partial x} + \frac{\partial v}{\partial y} = 0$$

for $x \in [-1, 1], y \in [-0.5, 0.5]$ and $t \in [0, 1]$ and with Dirichlet boundary conditions. As before, we consider $t \in [0, 0.5]$ as the interpolation domain and $t \in (0.5, 1]$ as the extrapolation domain.

## A.2  EFFECTS OF MODEL PARAMETERS ON EXTRAPOLATION PERFORMANCE

### A.2.1  BURGER'S EQUATION

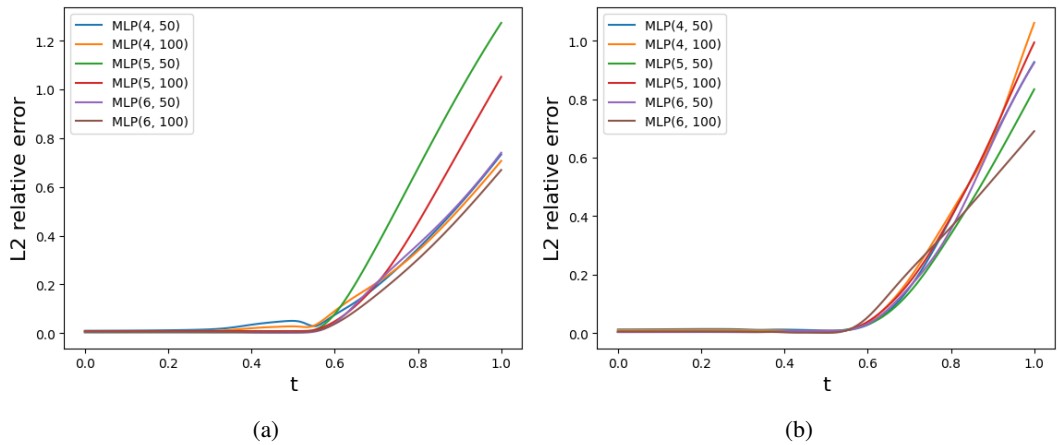

(a)                                                    (b)

Figure 6: $L^2$ relative extrapolation errors of various MLPs with $tanh$ activation in **(a)**, and with $sin$ activation in **(b)**. Trained on $[0, 0.5]$ using the same hyperparameters as in Section 3.1.

### A.2.2  ALLEN-CAHN EQUATION

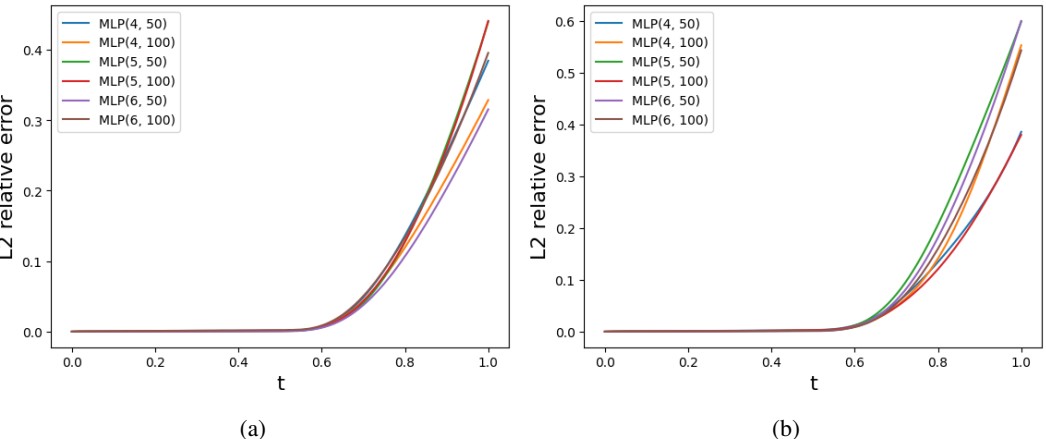

(a)                                                    (b)

Figure 7: $L^2$ relative extrapolation errors of various MLPs with $tanh$ activation in **(a)**, and with $sin$ activation in **(b)**. Trained on $[0, 0.5]$ using the same hyperparameters as in Section 3.1.

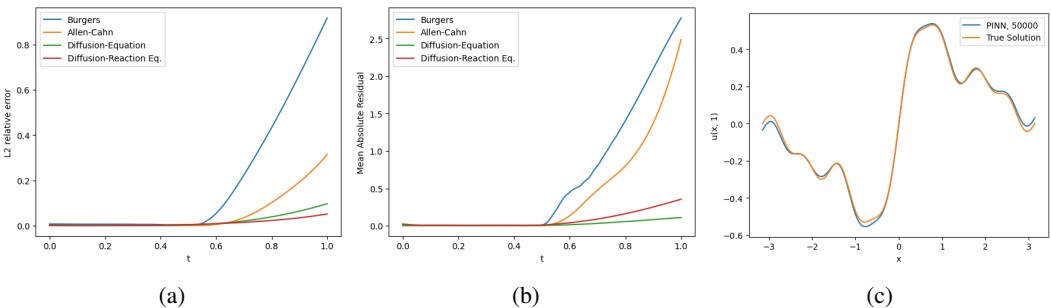

(a)              (b)              (c)

Figure 8: **(a)** $L^2$ relative extrapolation error of MLP(5, 64) with $tanh$ activation, trained on $[0, 0.5]$. **(b)** MAR for the same MLP, and **(c)** the solution for the diffusion-reaction equation at $t = 1$ and the function learned by the corresponding MLP.

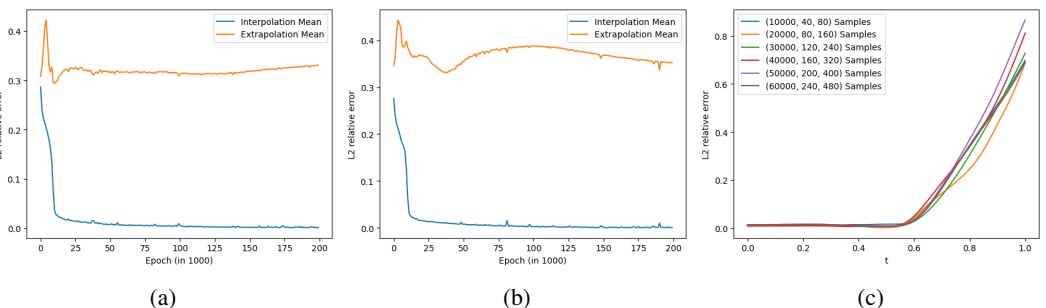

(a)              (b)              (c)

Figure 9: Mean $L^2$ relative errors over the interpolation (extrapolation) domain of MLP(5, 64) with $tanh$ activation **(a)** and with $\sin$ activation **(b)** with increasing number of training epochs trained to solve the Burger's equation. **(c)** plots the relative error against the number of samples, in the order (domain, boundary condition, initial condition).

## A.3 ANALYZING EXTRAPOLATION PERFORMANCE IN THE FOURIER DOMAIN

### A.3.1 ALLEN-CAHN EQUATION

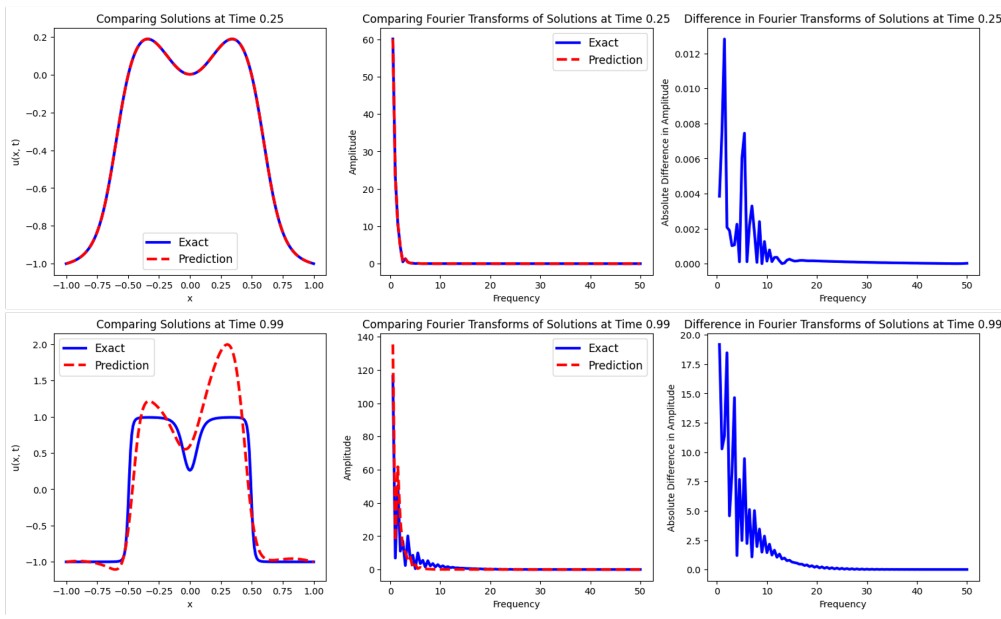

Figure 10: For times $t = 0.25$ (top, interpolation) and $t = 0.99$ (bottom, extrapolation), we plot the reference and predicted solutions in the spatio-temporal (left) and Fourier (middle) domains for the Allen-Cahn equation. The absolute difference in the Fourier spectra is plotted on the right.

### A.3.2 DIFFUSION EQUATION

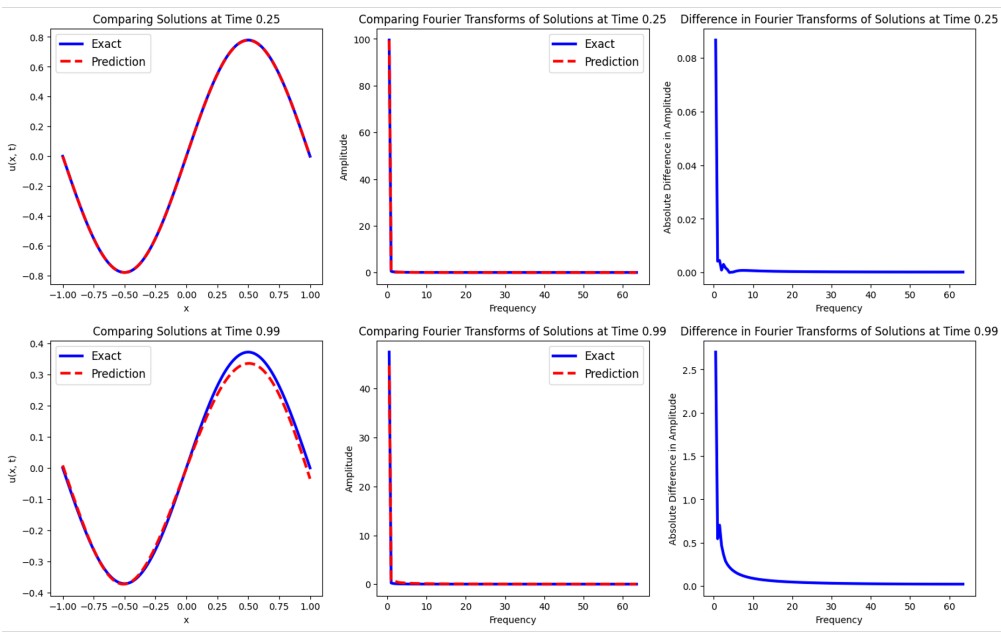

Figure 11: For times $t = 0.25$ (top, interpolation) and $t = 0.99$ (bottom, extrapolation), we plot the reference and predicted solutions in the spatio-temporal (left) and Fourier (middle) domains for the diffusion equation. The absolute difference in the Fourier spectra is plotted on the right.

### A.3.3 DIFFUSION-REACTION EQUATION

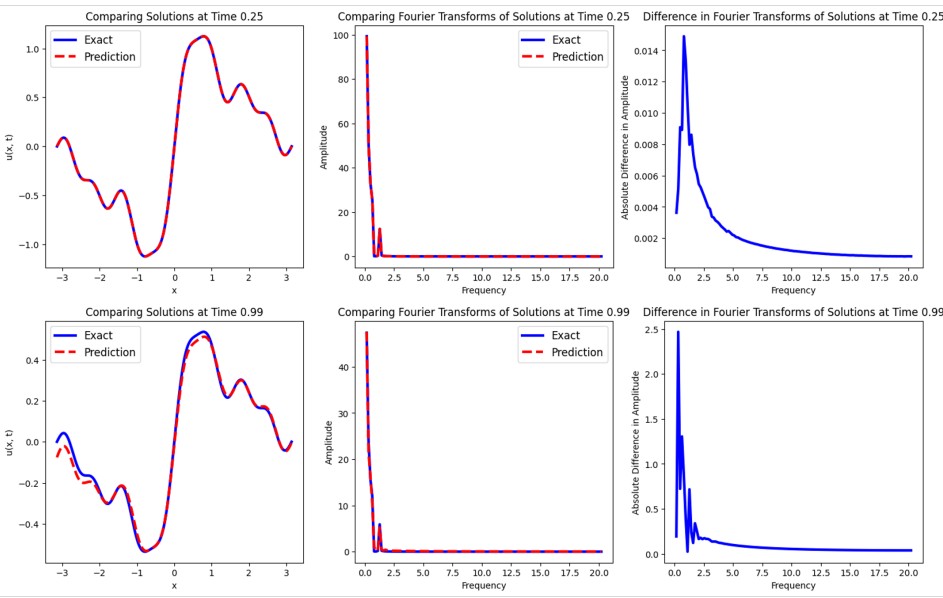

Figure 12: For times $t = 0.25$ (top, interpolation) and $t = 0.99$ (bottom, extrapolation), we plot the reference and predicted solutions in the spatio-temporal (left) and Fourier (middle) domains for the diffusion-reaction equation. The absolute difference in the Fourier spectra is plotted on the right.

### A.3.4 HEAT EQUATION

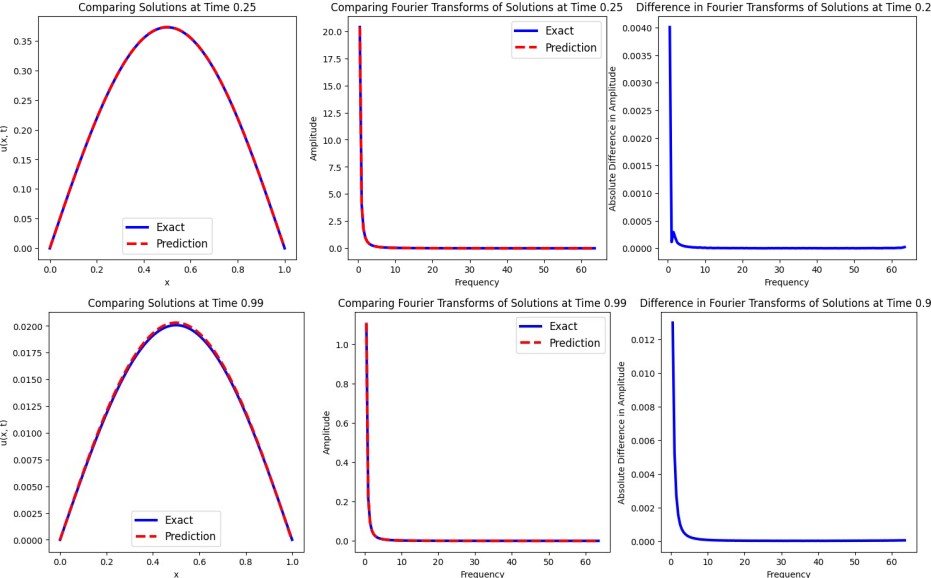

Figure 13: For times $t = 0.25$ (top, interpolation) and $t = 0.99$ (bottom, extrapolation), we plot the reference and predicted solutions in the spatio-temporal (left) and Fourier (middle) domains for the heat equation. The absolute difference in the Fourier spectra is plotted on the right.

### A.4 WASSERSTEIN-FOURIER DISTANCE PLOTS

For each of our four PDEs, we plot the pairwise Wasserstein-Fourier distances for both the reference and predicted solutions for $(t_1, t_2) \in \{0, 0.01, \ldots, 0.99\} \times \{0, 0.01, \ldots, 0.99\}$. We also plot the absolute difference between the two pairwise distance matrices to understand where the predicted solution is failing to capture the changing spectra. All differences are clipped below at $10^{-3}$ for stability reasons.

#### A.4.1 BURGERS' EQUATION

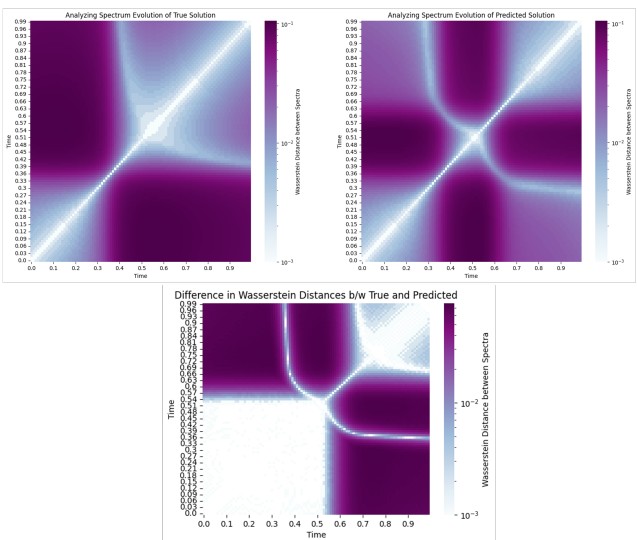

Figure 14: Pairwise Wasserstein-Fourier distances for the Burgers' equation. Reference solution (top left), predicted solution (top right), absolute difference (bottom).

#### A.4.2 ALLEN-CAHN EQUATION

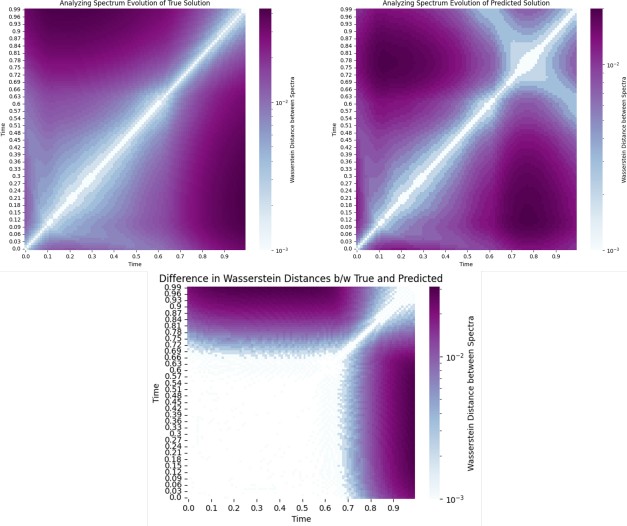

Figure 15: Pairwise Wasserstein-Fourier distances for the Allen-Cahn equation. Reference solution (top left), predicted solution (top right), absolute difference (bottom).

### A.4.3 DIFFUSION EQUATION

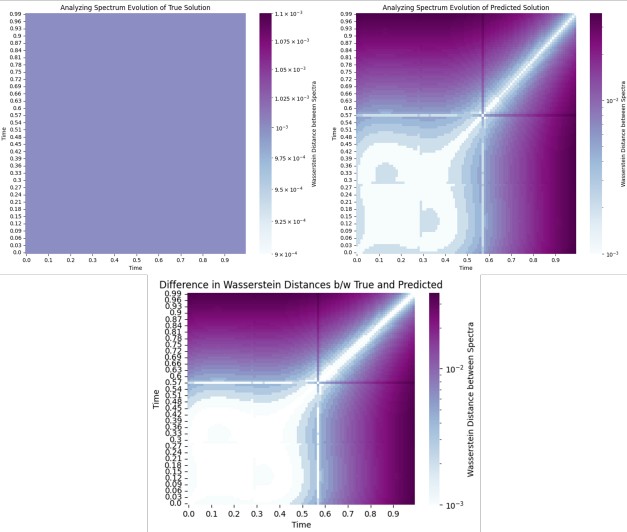

Figure 16: Pairwise Wasserstein-Fourier distances for the diffusion equation. Reference solution (top left), predicted solution (top right), absolute difference (bottom).

### A.4.4 DIFFUSION-REACTION EQUATION

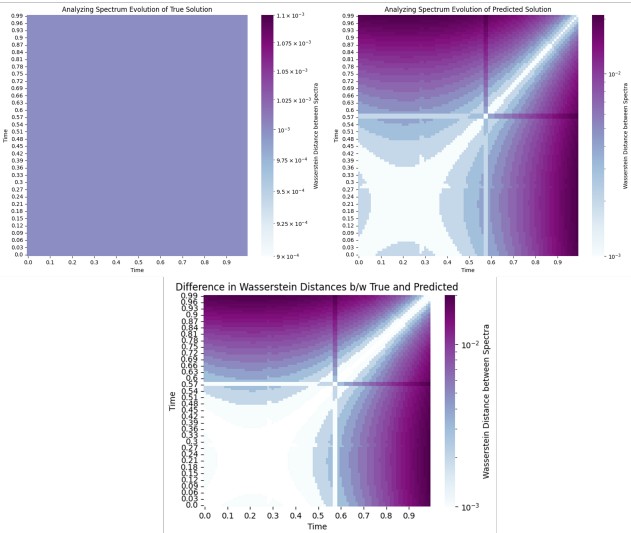

Figure 17: Pairwise Wasserstein-Fourier distances for the diffusion-reaction equation. Reference solution (top left), predicted solution (top right), absolute difference (bottom).

## A.5 EXPERIMENTS WITH UNCHANGING SUPPORT

Here, we examine the solutions for the PDEs examined in section 3.2 in the Fourier domain.

### A.5.1 VARYING THE SIZE OF THE SUPPORT

We first look at the PDE defined in equations (4) and (5). The reference solution is given by $u(x,t) = e^{-t}\left(\sum_{j=1}^{K}\frac{\sin(jx)}{j}\right)$. For a fixed $K$, the support of the Fourier spectra is constant – we plot the solutions for $K \in \{10, 15, 20\}$ in Figures 18, 19, and 20 respectively.

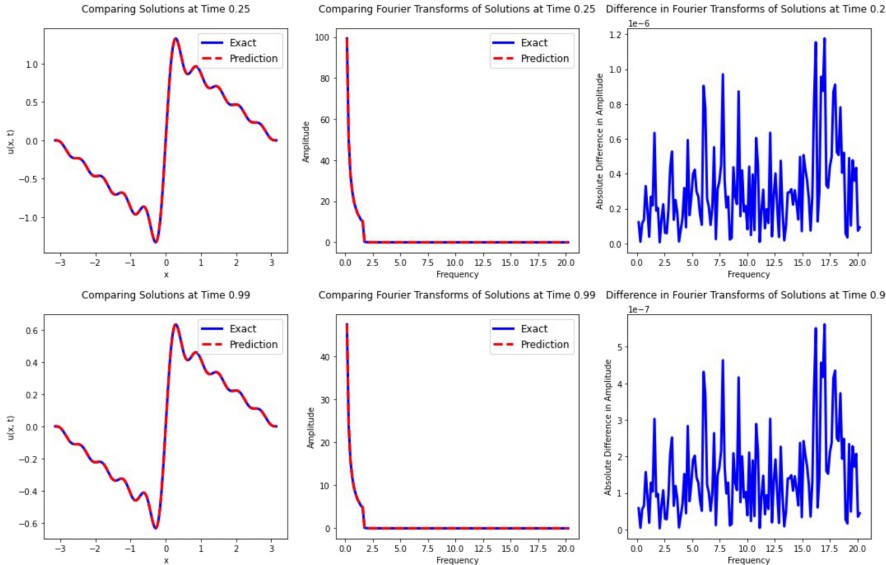

Figure 18: For times $t = 0.25$ (top, interpolation) and $t = 0.99$ (bottom, extrapolation), we plot the reference and predicted solutions in the spatio-temporal (left) and Fourier (middle) domains for $K = 10$. The absolute difference in the Fourier spectra is plotted on the right.

While extrapolation behavior is quite good, the highest frequency is still relatively small compared to the Burgers' or Allen-Cahn equations. To further examine whether spectral bias is a concern, we train a PINN on the PDE defined by

$$\frac{\partial u}{\partial t} = \frac{\partial^2 u}{\partial x^2} + e^{-t}\left(\sum_{j=1}^{k}\frac{(\pi j)^2 - 1}{j}\sin(\pi \cdot jx)\right)$$

with reference solution $u(x,t) = e^{-t}\left(\sum_{j=1}^{K}\frac{\sin(\pi \cdot jx)}{j}\right)$ for $K = 20$. The results are plotted in Figure 21. Note that the reference solution has frequencies as high as 10, similar to Allen-Cahn, but extrapolation remains near-perfect.

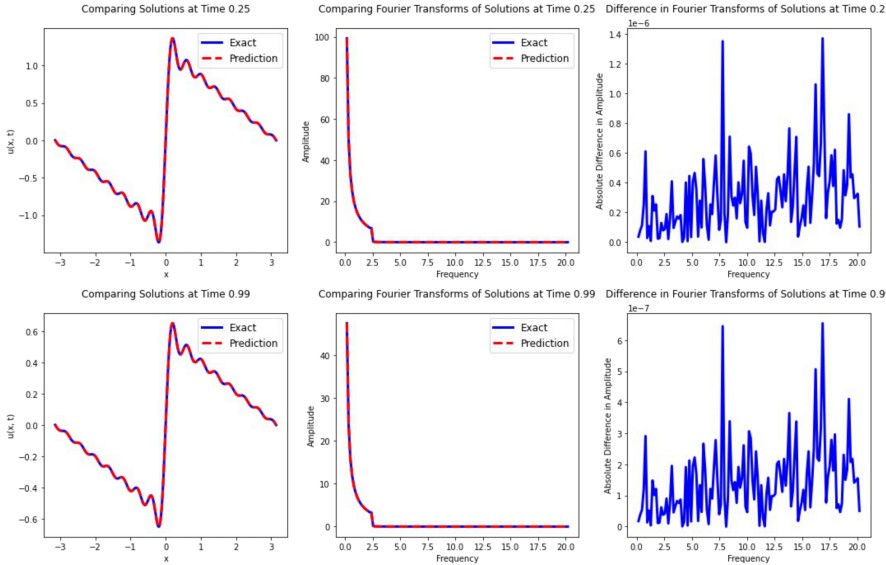

Figure 19: For times $t = 0.25$ (top, interpolation) and $t = 0.99$ (bottom, extrapolation), we plot the reference and predicted solutions in the spatio-temporal (left) and Fourier (middle) domains for $K = 15$. The absolute difference in the Fourier spectra is plotted on the right.

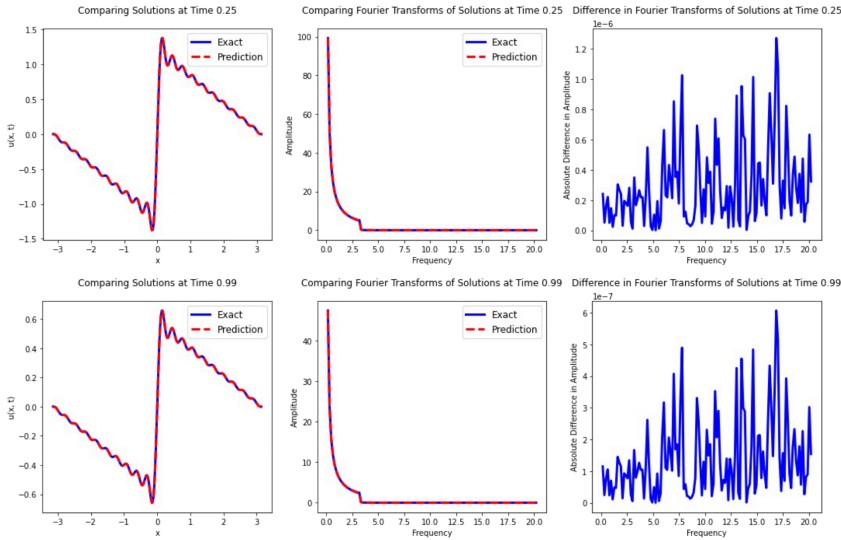

Figure 20: For times $t = 0.25$ (top, interpolation) and $t = 0.99$ (bottom, extrapolation), we plot the reference and predicted solutions in the spatio-temporal (left) and Fourier (middle) domains for $K = 20$. The absolute difference in the Fourier spectra is plotted on the right.

### A.5.2 VARYING AMPLITUDE DECAY

Next, we look at the PDE defined in equation (6) with reference solution $u(x,t) = e^{-Mt}\left(\sin(x) + \frac{\sin(2x)}{2} + \frac{\sin(3x)}{3} + \frac{\sin(4x)}{4} + \frac{\sin(8x)}{8}\right)$. For a fixed value of $M$, the support remains constant over time, but the amplitudes of the Fourier coefficients decay more rapidly over time for larger $M$. We plot the solutions for $M \in \{1, 3, 5.5\}$ in Figures 22, 23, and 24 respectively.

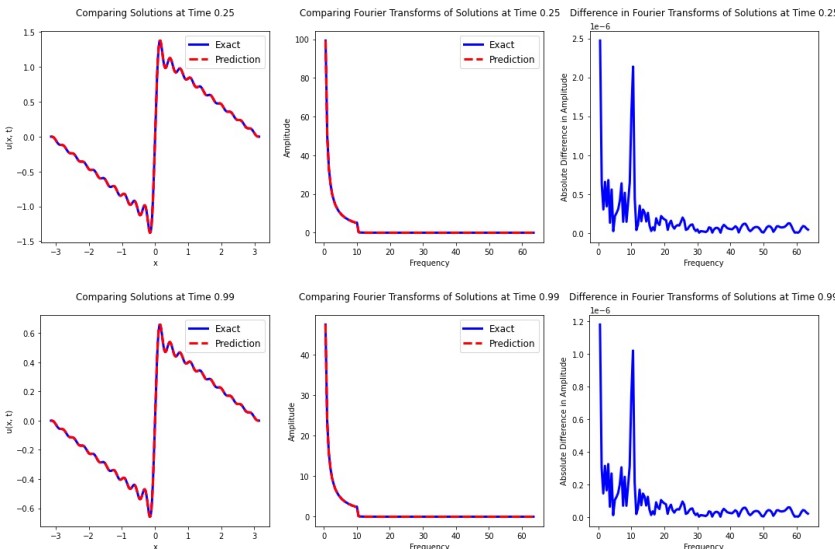

Figure 21: For times $t = 0.25$ (top, interpolation) and $t = 0.99$ (bottom, extrapolation), we plot the reference and predicted solutions in the spatio-temporal (left) and Fourier (middle) domains. The absolute difference in the Fourier spectra is plotted on the right.

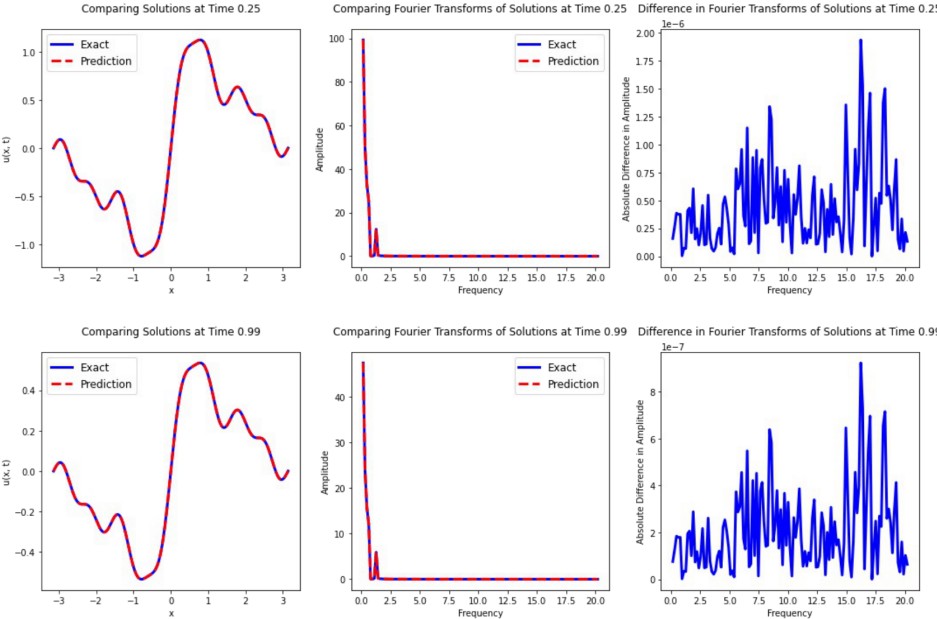

Figure 22: For times $t = 0.25$ (top, interpolation) and $t = 0.99$ (bottom, extrapolation), we plot the reference and predicted solutions in the spatio-temporal (left) and Fourier (middle) domains for $M = 1$. The absolute difference in the Fourier spectra is plotted on the right.

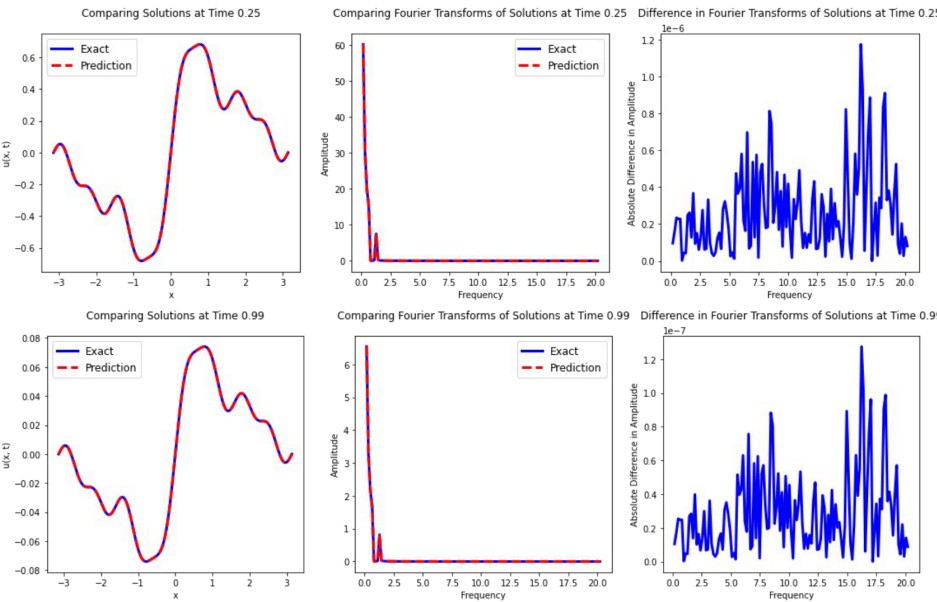

Figure 23: For times $t = 0.25$ (top, interpolation) and $t = 0.99$ (bottom, extrapolation), we plot the reference and predicted solutions in the spatio-temporal (left) and Fourier (middle) domains for $M = 3$. The absolute difference in the Fourier spectra is plotted on the right.

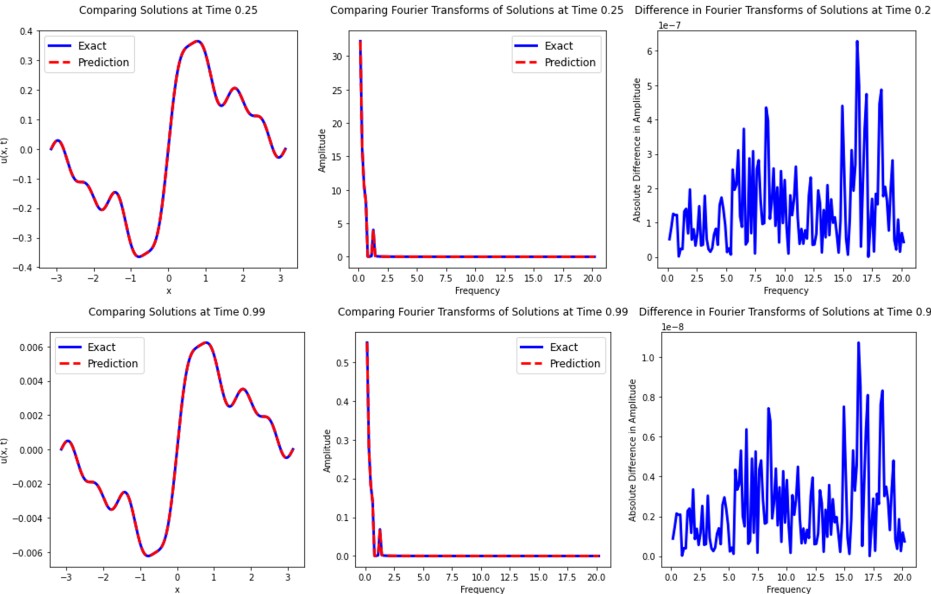

Figure 24: For times $t = 0.25$ (top, interpolation) and $t = 0.99$ (bottom, extrapolation), we plot the reference and predicted solutions in the spatio-temporal (left) and Fourier (middle) domains for $M = 5.5$. The absolute difference in the Fourier spectra is plotted on the right.

### A.6    PREDICTING EXTRAPOLATION BEHAVIOR THROUGH FOURIER REPRESENTATIONS

Given the observed relationship between Fourier spectra and extrapolation behavior, one may ask how well a reference solution's Fourier spectra predict the extrapolation performance of a PINN. Towards answering this question, we train a vanilla PINN – MLP(4, 50) – on the Burger's equation for 50 different viscosities, equally spaced from $\frac{0.001}{\pi}$ to $\frac{0.1}{\pi}$.

Given these learned solutions, we obtain relative L2 errors for each of the 50 PDEs, comparing them to reference solutions obtained via numerical methods. Finally, we train a simple 4-layer MLP to predict the relative L2 errors for $t = \{0, 0.1, \ldots, 1.0\}$ from the Fourier transforms of the reference solution at $t = \{0, 0.05, 0.1, \ldots, 1.0\}$ with standard MSE loss. For the purposes of numerical stability, we predict scaled errors, where all errors are multiplied by a factor of 10.

We withhold 5 of the 50 PDEs as a test set and evaluate the model's performance on this set. To ensure that we have a representative test set, we use stratified sampling to split our range of viscosities into five contiguous regions, sampling one PDE from each region for the test set. Predictions are shown in Figure 25 for these 5 PDEs, with the model achieving an $R^2$ of $0.85$.

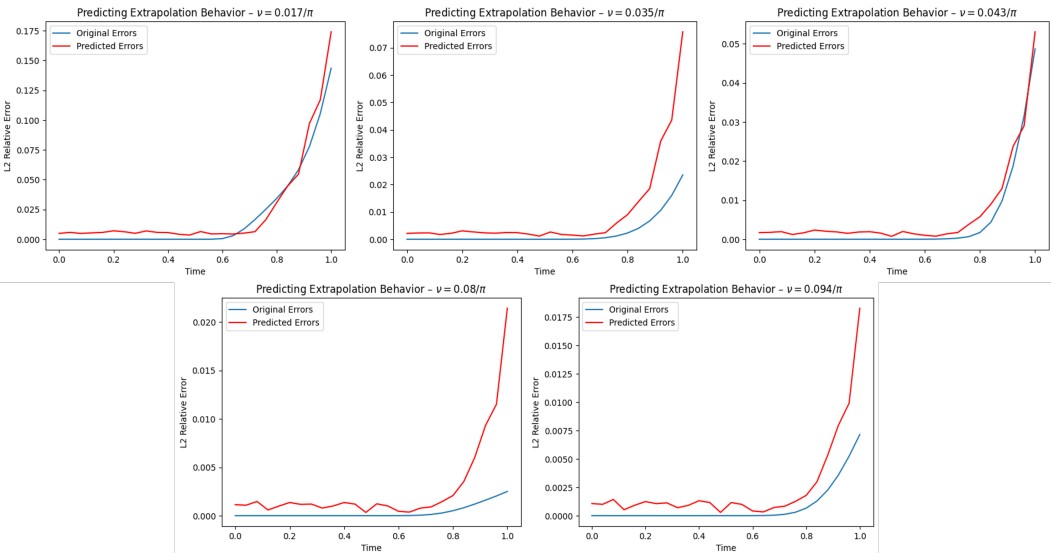

Figure 25: True and predicted L2 Relative Errors for the Burgers' PDEs withheld for the test set. Our model achieves an $R^2$ of $0.85$ on these 5 samples.

## A.7 MULTI-SCALE FOURIER FEATURE NETWORKS

| Sigmas | Domain Loss | Boundary Loss | Int. Error | Ext. Error | Int. MAR | Ext. MAR |
|--------|-------------|---------------|------------|------------|----------|----------|
| 1, 5 | 2.42e-4 | 2.95e-7 | 0.0026 | 0.7709 | 0.0057 | 1.6353 |
| 1, 10 | 5.35e-5 | 7.24e-7 | 0.0034 | 0.5612 | 0.0093 | 1.7226 |
| 1, 15 | 1.23e-4 | 3.07e-6 | 0.0272 | 0.5379 | 0.0289 | 2.0883 |
| 1, 5, 10 | 7.25e-5 | 5.19e-5 | 0.0156 | 0.7985 | 0.0127 | 2.2197 |
| No MFFN | 3.35e-5 | 1.66e-6 | 0.0031 | 0.5261 | 0.0082 | 1.1964 |

Table 3: Extrapolation performance of Multi-Fourier Feature Networks with 4 layers, 50 neurons each, and $sin$ activation trained on the Burger's equation specified in Appendix A.1, for various values of sigma. The last row provides the baseline comparison by using the standard architecture without multi-Fourier feature embeddings for the input.

## A.8 EFFECTS OF TRANSFER LEARNING ON INTERPOLATION & EXTRAPOLATION

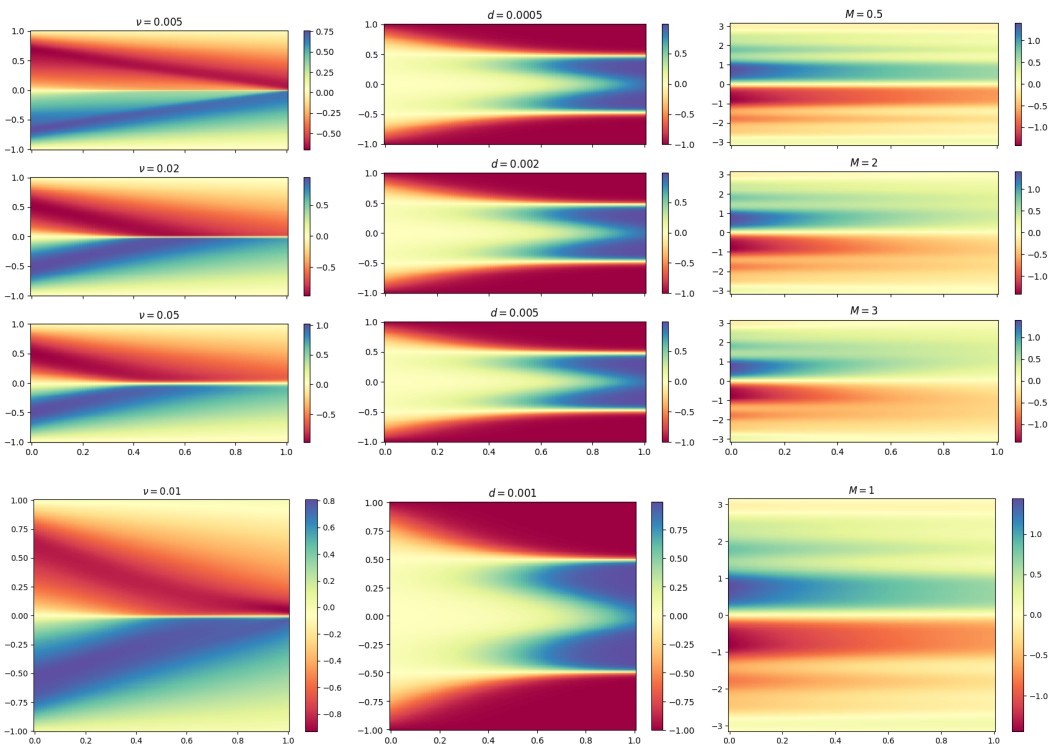

Figure 26: Solutions learned during transfer learning (top row) and solution learned for the target PDE (bottom row), for the Burger's equation **(left)**, the Allen-Cahn equation **(center)**, and the Diffusion-Reaction equation specified in Appendix A.1 **(right)**.

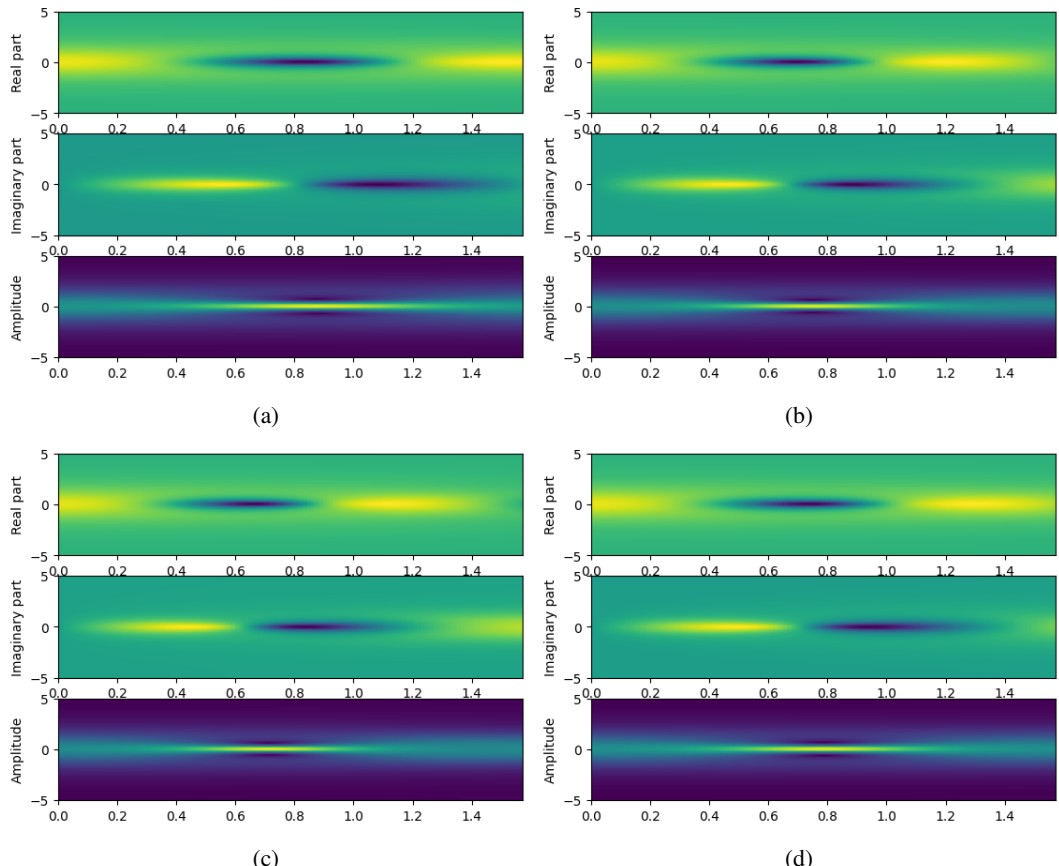

Figure 27: Solutions learned during transfer learning on the non-linear Schrodinger equation (a, b, c) and solution learned for the target PDE (d).

| Setting | Int. Domain Loss | Int. Boundary Loss | Int. Combined Loss |
|---------|------------------|--------------------|--------------------|
| Baseline | $0.00191 \pm 0.00202$ | $0.00134 \pm 0.00183$ | $0.00246 \pm 0.00225$ |
| Transfer $t \in [0, 0.5]$ | $0.00458 \pm 0.00178$ | $0.05612 \pm 0.12291$ | $0.08345 \pm 0.16787$ |
| Transfer $t \in [0, 1]$ | $0.03245 \pm 0.02273$ | $0.00348 \pm 0.00299$ | $0.03296 \pm 0.02333$ |

Table 4: Interpolation loss terms for the baseline setting (no transfer learning), transfer learning from half the domain ($t \in [0, 0.05]$), and transfer learning from the full domain ($t \in [0, 1]$), in the form mean $\pm$ std. Values obtained from 15 MLPs per setting, trained on the Burger's equation.

| Setting | Ext. Domain Loss | Ext. Boundary Loss | Ext. Combined Loss |
|---------|------------------|--------------------|--------------------|
| Baseline | $11.6506 \pm 6.58194$ | $0.00055 \pm 0.00038$ | $9.93962 \pm 5.42231$ |
| Transfer $t \in [0, 0.5]$ | $3.49251 \pm 2.59067$ | $0.04385 \pm 0.09974$ | $3.75796 \pm 2.70572$ |
| Transfer $t \in [0, 1]$ | $0.45959 \pm 0.44864$ | $0.00397 \pm 0.00337$ | $0.52353 \pm 0.35683$ |

Table 5: Extrapolation loss terms for the baseline setting (no transfer learning), transfer learning from half the domain ($t \in [0, 0.05]$), and transfer learning from the full domain ($t \in [0, 1]$), in the form mean $\pm$ std. Values obtained from 15 MLPs per setting, trained on the Burger's equation.

## A.9 INVESTIGATIONS INTO DYNAMIC PULLING

We examine the improved extrapolation performance of the dynamic pulling method (DPM) proposed by Kim et al. (2020). In brief, their method modifies the gradient update in PINN training

to dynamically place more emphasis on decreasing the domain loss in order to stabilize the domain loss curve during training.

We implement DPM for the Burgers' equation with viscosity $\nu = \frac{0.01}{\pi}$ and compare to a standard PINN without DPM. For both sets of experiments, we use the architecture that Kim et al. (2020) found to have the best extrapolation performance on this particular PDE (MLP PINN with residual connections, 8 hidden layers, 20 hidden units per layer, tanh activation, and Xavier normal initialization). We train using Adam with learning rate $0.005$ and otherwise default parameters. When training with DPM, we use $\epsilon = 0.001$, $\Delta = 0.08$, $w = 1.001$.

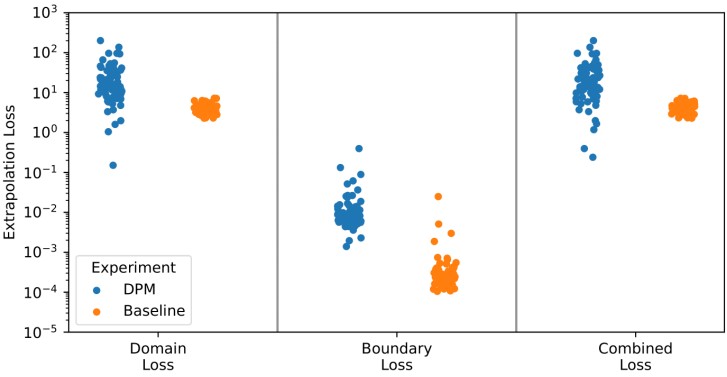

Figure 28: Domain, boundary, and combined mean squared extrapolation error between our baseline (PINNs trained from scratch) and PINNs with DPM-modified gradient updates. We train 77 models with DPM and 60 models without DPM. The only difference between model runs is the random seed.

We train 77 DPM models and 60 standard models, differing only in the random seed. Our results are shown in Figure 28. As before, we find that our extrapolation error is dominated by the domain loss. Notably, we find that DPM on average does considerably worse in extrapolation than our baseline. However, the errors are higher variance and a number of DPM models perform better in extrapolation than any of our baseline models. The particular training dynamics induced by DPM which cause these shifts are unclear but potentially deserve more detailed investigation.

## A.10 TRAINING & HARDWARE DETAILS

| Section | Model | Activation | Initialization | Optimizer | LR | Epochs | Samples |
|---|---|---|---|---|---|---|---|
| 3.1 (Figure 1) | MLP(4, 50) | $tanh$ | Xavier | Adam | 1e-4 | 50000 | 10000, 40, 80 |
| 3.2 (Figure 2) | MLP(4, 50) | $tanh$ (a), $sin$ (b) | Xavier | Adam | 1e-4 | Varying | Varying |
| 4.1 (Figure 3) | MLP(3, 20) | $tanh$ | Xavier | Adam | 1e-4 | 50000 | 10000, 40, 80 |
| 4.2 (Figure 4) | MLP(6, 50) | $tanh$ | Xavier | Adam | 1e-4 | 100000 | 20000, 80, 160 |
| 5 (Figure 5) | MLP(5, 100) | $tanh$ | Xavier | Adam | 1e-4 | Varying | Varying |

Table 6: Training details for the experiments presented in the main text. Here, MLP(4, 50) refers to a fully-connected neural network with 4 layers and 50 neurons per layer; Xavier refers to the Xavier normal initialization; Adam refers to the Adam optimizer with all parameters set to default; and the samples are in the form (domain, boundary condition, initial condition).

**Hardware:** All our experiments were conducted on an NVIDIA A100 GPU with 16 GB RAM.

