# OpenReview forum: "Understanding and Mitigating Extrapolation Failures in Physics-Informed Neural Networks"
_ICLR.cc/2024/Conference — Submitted to ICLR 2024_

### Official Review · Reviewer_dasc · 2023-10-27

**Soundness:** 3 good
**Presentation:** 3 good
**Contribution:** 2 fair
**Rating:** 5
**Confidence:** 4

**Summary:**

The paper studies the extrapolation of PINNs based on the Fourier spectrum shifts.

**Strengths:**

The author analyzes the extrapolation performance of PINNs based on the Weighted Wasserstein-Fourier distance (WWF) in different time domains during training and testing.

**Weaknesses:**

The extrapolation problem of PINN is not a significant problem, as we can always finetune the PINN if long-time prediction is needed, or we can train a new PINN on the new domain.

In the context of domain adaptation and domain generalization in computer vision, the conclusion of this paper does not seem to be novel. The distance between Fourier components during train & test domains is just like the concept of distribution shift in computer vision, where people use the KL divergence and other metrics to quantify the representation distribution between train and test to predict the out-of-domain  /out-of-distribution generalization performance.

From this viewpoint, we can also reinterpret the authors' conclusion: We find that failure to extrapolate is not caused by high frequencies in the solution function, but rather by shifts in the support of the Fourier spectrum over time.

The so-called "shifts in the support of the Fourier spectrum over time" is just the distribution shift in computer vision.

And one can actually derive a rigorous mathematical bound for the out-of-domain PINN error.

**Questions:**

Please justify the importance of PINN's extrapolation: why don't we just train a new model/finetune?
Please explain your novelty over the concept of domain adaptation and domain generalization in computer vision.

---

> ### Author Response · Authors · 2023-11-16
>
> We thank the reviewer for their feedback and for their very interesting suggestions.
>
> > Please justify the importance of PINN's extrapolation: why don't we just train a new model/finetune?
>
> As remarked on by an increasing number of papers [1, 2, 3], models that can extrapolate well are generally desirable, for example because extrapolation failures indicate that the model hasn’t correctly learned the underlying physical laws [1], which are the reason for using PINNs in the first place. Simply training a new model (on a larger domain) does not address this issue as the new model will still suffer from poor extrapolation outside of its training domain.
>
> > In the context of domain adaptation and domain generalization in computer vision, the conclusion of this paper does not seem to be novel. The distance between Fourier components during train & test domains is just like the concept of distribution shift in computer vision, where people use the KL divergence and other metrics to quantify the representation distribution between train and test to predict the out-of-domain /out-of-distribution generalization performance. From this viewpoint, we can also reinterpret the authors' conclusion: We find that failure to extrapolate is not caused by high frequencies in the solution function, but rather by shifts in the support of the Fourier spectrum over time. The so-called "shifts in the support of the Fourier spectrum over time" is just the distribution shift in computer vision. And one can actually derive a rigorous mathematical bound for the out-of-domain PINN error.
>
> Unfortunately, deriving generalization bounds for PINNs has been found to be significantly harder than to simply adapt concepts from the domain adaptation and domain generalization literature [4]. In particular, minimizing the PDE residual in PINNs does not straightforwardly control the generalization error [2], which makes the PINN case different from, for example, computer vision.
>
> In addition, the theoretical results derived so far for PINN performance outside of the training data are limited to generalization errors, where we are interested in the error committed on points sampled from within the convex hull of the training data. There is, to the best of our knowledge, no theoretical work on the extrapolation performance of PINNs, where the data of interest is far away from the training domain.
>
> > Please explain your novelty over the concept of domain adaptation and domain generalization in computer vision.
>
> Please see our comment above. We agree with the reviewer that it would be desirable to have results for PINNs that are similar to the domain adaptation literature in, for example, computer vision. We are aware of at least one work that uses tools from domain adaptation to at least empirically improve the extrapolation performance of PINNs [1]. As mentioned in our discussion section, we think that a theoretical investigation of the extrapolation performance of PINNs would be a logical next step, and we believe that our experimental results point into a potentially promising direction, i.e. spectral shifts.
>
> [1] Jungeun Kim, Kookjin Lee, Dongeun Lee, Sheo Yon Jin, and Noseong Park. Dpm: A novel training method for physics-informed neural networks in extrapolation, 2020.
>
> [2] Taniya Kapoor, Abhishek Chandra, Daniel M Tartakovsky, Hongrui Wang, Alfredo Nunez, and Rolf Dollevoet. Neural oscillators for generalization of physics-informed machine learning. arXiv preprint arXiv:2308.08989, 2023.
>
> [3] Andrea Bonfanti, Roberto Santana, Marco Ellero, and Babak Gholami. On the hyperparameters influencing a pinn’s generalization beyond the training domain, 2023.
>
> [4] Siddhartha Mishra and Roberto Molinaro. Estimates on the generalization error of physics-informed neural networks for approximating a class of inverse problems for pdes. IMA Journal of Numerical Analysis, 42(2):981–1022, 2022.

---

> > ### Author Response · Authors · 2023-11-22
> >
> > We’d like to thank the reviewer again for the comments and hope that our rebuttal has addressed their concerns/questions. If there is anything that we can clarify/answer before the discussion period ends, we would be happy to do so.

---

### Official Review · Reviewer_SeWR · 2023-10-28

**Soundness:** 2 fair
**Presentation:** 2 fair
**Contribution:** 2 fair
**Rating:** 5
**Confidence:** 4

**Summary:**

The paper introduces extrapolation failures of PINNs, which studies the out-of-domain behavior of PINNs. The paper then analyzes the extrapolation failures in the scope of Weighted Wasserstein-Fourier Distance and spectral bias, showing the PDEs with blockwise WWFs tend to have extrapolation failures, and when extrapolation happens, the prediction spectral shifts from the true solution.

The paper then proposes a transfer learning strategy that effectively mitigates extrapolation failures.

**Strengths:**

1. The paper introduces extrapolation failures of PINNs, which studies the out-of-domain behavior of PINNs and seems to be a novel and promising research topic.
2. The paper leverages a novel scope, Weighted Wasserstein-Fourier Distance, to analyze spectral shifts for extrapolation failures. The analysis shows that (1) PDEs with blockwise WWFs tend to have extrapolation failures and (2) when extrapolation happens, the prediction spectral shifts from the true solution.
3. The paper proposes a transfer learning strategy that mitigates extrapolation failures.

**Weaknesses:**

1. The theory of this paper is not solidly developed enough. The conclusion of the paper, says spectral shift, is drawn from observations of several specific types of PDEs, can be empiricism, and may not be generalizable enough for other PDEs. In addition, justifying whether a PDE will suffer from an extrapolation failure using WWF can be difficult in practice. To examine whether the WWF is blockwise or not, it requires true $f_s$ for $s\in I$ and $f_t$ for $t\in E$, while $f_s$ and $f_t$ are not available for most practical cases.

2. Leveraging the concept of spectral bias for extrapolation as a hypothesis is problematic. The spectral bias states that NN tends to learn low-frequency components more easily and faster than high-dimensional components during training [1]. While extrapolation is a validation/testing process. Thus, one should not explain a phenomenon in testing with a theory for training, or use it as a hypothesis (despite that the paper beats the hypothesis to the end). The hypothesis reference paper [2] does not make any statement on extrapolation with spectral bias either, they only assert PINNs fail to train when the training time window becomes large, possibly due to spectral bias.

3. The presentation of the paper can be improved, say most figures can be denser to save space, and most tables can have nicer borders, etc.

[1]. Rahaman, Nasim, et al. "On the spectral bias of neural networks." International Conference on Machine Learning. PMLR, 2019.

[2]. Wang, Sifan, and Paris Perdikaris. "Long-time integration of parametric evolution equations with physics-informed deeponets." Journal of Computational Physics 475 (2023): 111855.

**Questions:**

1. For the proposed transfer learning method, what is the benefit of transfer learning rather than directly learning new PINNs over the full domain?  As shown by massive previous works, PINNs can easily learn accurate solutions for Burger's equation, as showcased in this work.

2. Could the author explain why (or what does it mean) for a constant WWF distance of the true solutions for diffusion/reaction-diffusion equation in Figures 15 & 16?

3. Could the author explain why diffusion and reaction-diffusion (Figures 15 & 16) show similar WWF distance differences between true solutions and predictions as Burger's and Allen-Cahn (Figures 13 & 14), but the first two do not show significant extrapolation failure, while the latter two show significant extrapolation failures (Figure 7)?

---

> ### Author Response · Authors · 2023-11-16
>
> We thank the reviewer for their feedback and for their very useful comments.
>
> > The theory of this paper is not solidly developed enough. The conclusion of the paper, says spectral shift, is drawn from observations of several specific types of PDEs, can be empiricism, and may not be generalizable enough for other PDEs. In addition, justifying whether a PDE will suffer from an extrapolation failure using WWF can be difficult in practice.
>
> In this paper, we focused primarily on putting forth evidence supporting the novel hypothesis that spectral shifts could be responsible for the poor extrapolation behavior of PINNs. We certainly welcome a more thorough theoretical investigation of this phenomenon, but we chose to leave that to future work.
>
> We would also like to point out that our focus here is not on practical applications just yet, but on acquiring a basic understanding of what can make PINNs fail to extrapolate. Future work should extend our analysis to state-of-the-art PINN models and, as the reviewer pointed out, investigate if the WWF can be used in practical applications.
>
> > Leveraging the concept of spectral bias for extrapolation as a hypothesis is problematic. The spectral bias states that NN tends to learn low-frequency components more easily and faster than high-dimensional components during training [1]. While extrapolation is a validation/testing process. Thus, one should not explain a phenomenon in testing with a theory for training, or use it as a hypothesis (despite that the paper beats the hypothesis to the end).
>
> We thank the reviewer for voicing their concern. The issue with the citation pointed out by the reviewer has been fixed. By spectral bias, we indeed mean PINNs tendency to learn lower frequencies better and faster in interpolation. However, prior work has also studied the connections between NTK theory, spectral bias, and extrapolation, for example [1]. Following this, we would expect most of the extrapolation error to come from the higher frequencies (the predicted function might become smooth or flat, as in [2]). Our Fourier-based results in section 3 show that this is not the case: a large share of the error seems to come from the lower frequencies.
>
> > The presentation of the paper can be improved, say most figures can be denser to save space, and most tables can have nicer borders, etc.
>
> We have generally made the representation a bit denser and have also moved some additional figures from the appendix to the main text.
>
> > For the proposed transfer learning method, what is the benefit of transfer learning rather than directly learning new PINNs over the full domain? As shown by massive previous works, PINNs can easily learn accurate solutions for Burger's equation, as showcased in this work.
>
> We agree that it is generally possible to train a new model on a larger domain (assuming that domain isn’t too large) to get accurate solutions. This is, however, based on the fact that PINNs can perform well in interpolation.
>
> As recent works have argued [3], models that can extrapolate well are generally desirable, for example because extrapolation failures indicate that the model hasn’t correctly learned the underlying physical laws [4], which are the reason for using PINNs in the first place.
> Simply training a new model (on a larger domain) does not address this issue as the new model will still suffer from poor extrapolation outside of its training domain.
>
> Transfer learning is interesting in this regard in that it seems to encourage inductive biases in the model that improve extrapolation performance (see our discussion in section 4) without training on a larger domain with the target PDE.
>
> [1] Xu, Keyulu, Mozhi Zhang, Jingling Li, Simon S. Du, Ken-ichi Kawarabayashi, and Stefanie Jegelka. "How neural networks extrapolate: From feedforward to graph neural networks." arXiv preprint arXiv:2009.11848 (2020).
>
> [2] Andrea Bonfanti, Roberto Santana, Marco Ellero, and Babak Gholami. On the hyperparameters influencing a pinn’s generalization beyond the training domain, 2023.
>
> [3] Taniya Kapoor, Abhishek Chandra, Daniel M Tartakovsky, Hongrui Wang, Alfredo Nunez, and Rolf Dollevoet. Neural oscillators for generalization of physics-informed machine learning. arXiv preprint arXiv:2308.08989, 2023.
>
> [4] Jungeun Kim, Kookjin Lee, Dongeun Lee, Sheo Yon Jin, and Noseong Park. Dpm: A novel training method for physics-informed neural networks in extrapolation, 2020.

---

> ### Author Response · Authors · 2023-11-16
>
> > Could the author explain why (or what does it mean) for a constant WWF distance of the true solutions for diffusion/reaction-diffusion equation in Figures 15 & 16?
>
> For numerical stability reasons, the WWF distance was clipped at $10^{-3}$ – the true WWF distance is in fact zero for both the diffusion and diffusion-reaction equations.
>
> This is because the support of the Fourier spectrum of the diffusion and diffusion-reaction equation does not change over time, only the amplitudes decrease. These changes in amplitude are not captured by the WWF, since we normalize the Fourier spectra before computing the Wasserstein Distance.
>
> Apologies for any confusion this may have caused – we have clarified this in our revision.
>
> > Could the author explain why diffusion and reaction-diffusion (Figures 15 & 16) show similar WWF distance differences between true solutions and predictions as Burger's and Allen-Cahn (Figures 13 & 14), but the first two do not show significant extrapolation failure, while the latter two show significant extrapolation failures (Figure 7)?
>
> Note that we do not claim that the WWF between true solutions and predictions is indicative of the size of the extrapolation error. We only claim this (and experimentally show it) for the WWF between the true solution in the interpolation region and in the extrapolation region.
>
> We believe that what the reviewer has observed here relates back to the previous comment, i.e. that the WWF is insensitive to differences in amplitude. The WWF (and WF more generally) only measures the change in the underlying distribution of different frequencies, neglecting any shift in overall amplitude. For example, multiplying the spectrum by a constant would lead to large L2 error, but the WWF distance would still be zero.
>
> As can be seen in section A.3 of the appendix, both Burger’s and Allen-Cahn exhibit significantly larger magnitude differences between predictions and true solutions in the Fourier spectra than both the diffusion and diffusion-reaction questions, only some of which is captured in the WWF distance.

---

> > ### Comment · Reviewer_SeWR · 2023-11-20
> > **Response to authors' comment**
> >
> > I thank the author for the explanation of my questions.
> >
> > A more general question here is whether the PDEs with extrapolation failures can be generalized simply based on the PDEs' types, say elliptic, parabolic, or hyperbolic.
> >
> > Also, would the author re-emphasize the three most important contributions of this work, instead of listing 5 contributions in the original paper?

---

> ### Author Response · Authors · 2023-11-20
>
> We thank the reviewer for going through our rebuttal.
>
> > Also, would the author re-emphasize the three most important contributions of this work, instead of listing 5 contributions in the original paper?
>
> We believe that the three most important contributions in our work are 1) PINNs can indeed extrapolate well for certain PDEs, even when high frequencies are present in the solution function, 2) PINNs’ extrapolation performance is linked to spectral shifts in the solution, and 3) transfer learning is beneficial for extrapolation performance when spectral shifts are present.
>
> We would be happy to adjust the section on novel contributions if the reviewer considers this shorter version to be more accessible.
>
> > A more general question here is whether the PDEs with extrapolation failures can be generalized simply based on the PDEs' types, say elliptic, parabolic, or hyperbolic.
>
> Unfortunately, based on our experiments, there doesn't seem to be an obvious connection between PDE type and PINN extrapolation performance: extrapolation works well on the heat equation (parabolic), but produces large errors on the Allen-Cahn equation (also parabolic, but non-linear). We believe that this is an interesting question, however, that might become important when deriving rigorous bounds on the extrapolation error.

---

> > ### Author Response · Authors · 2023-11-22
> >
> > We’d like to thank the reviewer again for the comments and hope that our most recent response has addressed their concerns/questions. If there is anything that we can clarify/answer before the discussion period ends, we would be happy to do so.

---

### Official Review · Reviewer_qKGX · 2023-10-29

**Soundness:** 3 good
**Presentation:** 4 excellent
**Contribution:** 3 good
**Rating:** 8
**Confidence:** 5

**Summary:**

The manuscript looks at extrapolation capabilities of the PINN solutions for multipl. The authors introduce a concept of spectral shifts that can be used to predict the PINN extrapolation performance.

The spectral shifts are used to analyse what features are not affecting the extrapolation performance using a set of seven different PDEs. Their results indicate that the number of layers, number of neurons , activation function, number of samples or training do not contribute.

To remedy the extrapolation challenge, the authors find that  transfer learning using similar PDEs, decreases significantly the extrapolation errors.

**Strengths:**

A significant analysis on the extrapolation capability of the  PINN in its basic structure.
Introduction of the weighted Wasserstein-Fourier distance as a measure.
Excellent and clear presentation of the results.

**Weaknesses:**

One may assume that extrapolation will be successful when the training data contains the types of behaviour that will occur in the extrapolated portion of the time domain. This would explain how the PINN solutions of PDEs without or with small spectral shift are viable for extrapolation. If the characteristics of the PDE solutions change drastically after some time scale i.e. have a large spectral shift.

I am missing an analysis where the a particular differential equation have short time behaviour which is spectrally "stable" and long term behaviour, where the spectral shift emerges.  Hence, the difference in the PDE behaviour may also come from how long a certain trajectory is followed, not intrinsically from what kind of equation it is. Then, a user of the method may feel safe to extrapolate a given PDF that has seemed to be "safe" but enter in the dangerous domain without warning.

It seems that safe way would be to use an alternative method to check the accuracy of the solution always, which makes the extrapolation capability less useful.

**Questions:**

Please, consider the possible issue I raised in the weaknesses part.

---

> ### Author Response · Authors · 2023-11-16
>
> We thank the reviewer for their detailed read of our submission and for the very encouraging feedback.
>
> We agree that experiments with a PDE that has a stable spectrum in the short term and a spectral shift in the long term would be very interesting. We are, unfortunately, not aware of an obvious PDE choice that exhibits this particular behavior, but would be very grateful for any suggestions.

---

> ### Comment · Reviewer_qKGX · 2023-11-21
>
> For example,  if a massive object travels across the solar system it can affect the orbits of the planets in a secular way.  From data of the planets before the crossing one cannot see that this would happen, until the object is already close. It is very hard  to extrapolate this based on data available before the encounter.  Also, check https://arxiv.org/pdf/1510.00591.pdf, this system does not even need a "rogue" hidden planet out there.
>
> Also, for example in materials sciences fatigue caused cracking  is also hard to extrapolate (without prior experiments covering the cracking timescale of the materials)  and leads to frequency shifts in the behaviour after a long period of contained frequency response.
>
> What I mean to say is that extrapolation outside the experimental data domain is always risky.

---

> ### Author Response · Authors · 2023-11-23
>
> We’d like to thank the reviewer for bringing our attention to these examples, and we agree that there may indeed be cases where spectral shifts emerge outside of the training domain and as a result may not be easily detectable.
>
> The focus of this paper is not necessarily on practical usage of the WWF metric; instead, we focused on better understanding precisely what features of a PDE made it easy/hard for PINNs to extrapolate. There are certainly cases where spectral shifts may be present in the original training domain and may signal potential extrapolation failures.
>
> However, we agree that there may be instances where simply examining the WWF/spectral dynamics in the training domain may fail to detect potentially poor extrapolation behavior due to later spectral shifts and where extrapolation is inherently uncertain/difficult — for example, in the cases the reviewer mentioned above.
>
> We believe these are valuable questions, and we thank the reviewer for their feedback and comments. We would certainly welcome a more thorough investigation of how one could better predict these spectral shifts from a limited domain in future work.

---

### Official Review · Reviewer_uE7H · 2023-11-01

**Soundness:** 2 fair
**Presentation:** 3 good
**Contribution:** 2 fair
**Rating:** 6
**Confidence:** 4

**Summary:**

The paper debunks the idea that the poor extrapolation performance of PINNs are due to the presence of high frequency components. The paper demonstrates the poor extrapolation performance is due to a shift in the support of the Fourier spectrum, which they refer to as the spectral shift. The paper introduces a metric to quantify this shift using Weighted Wasserstein Fourier distance (WWF). The paper finally shows that a transfer-learning based technique which trains a multi-headed PINN can be effective in providing better extrapolation performances.

**Strengths:**

1. The paper is overall well-written and easy to follow.
2. Analyzing the poor extrapolation performance using spectral shifts is novel and interesting.
3. The proposed WWF metric can be a good evaluation tool to visualize the spectral shifts.

**Weaknesses:**

1. Although the paper provides some empirical evidence of correlations of spectral shifts and poor extrapolation performance, the paper lacks a theoretical understanding of why the hypothesized Spectral Shift is the root cause of extrapolation error.
2. The motivation behind the proposed transfer learning approach is not clear. I would highly appreciate it if the authors can provide the connections between the observed spectral shifts and how transfer learning can mitigate it.
3. The empirical results for improved extrapolation performance are not convincing. The L2 Extrapolation Error on the Schrodinger Equation (imag) for the proposed method is still quite poor (290%), which is still quite unusable from a practical stand-point.

**Questions:**

**Comments/Questions:**
1. I understand the space constraints but figures 7a and b are quite important to demonstrate the results shown in Section 3.1.
2. The placement of Figure 1 can be improved. It is referenced in Page 6 while the Figure is present in page 1. Changing the location of the figure to a more appropriate location can improve the readability of the paper.
3. In my opinion, transfer Learning on the full domain is an unfair comparison, as the PDEs with a similar set of coefficients were already trained on the entire domain.

**Minor comments:**
The authors use a different citation format than the standard ICLR format. Using the original citation format would result in overflow of the text outside the page limit since they are considerably longer.

---

> ### Author Response · Authors · 2023-11-16
>
> We would like to thank the reviewer for their encouraging feedback and useful comments.
>
> > Although the paper provides some empirical evidence of correlations of spectral shifts and poor extrapolation performance, the paper lacks a theoretical understanding of why the hypothesized Spectral Shift is the root cause of extrapolation error.
>
> We agree that a theoretical analysis of the role of spectral shifts in the extrapolation performance of PINNs is a natural next step, and we included this as a promising future direction in our discussion section.
>
> In this paper, we focused primarily on putting forth evidence supporting the novel hypothesis that spectral shifts could be responsible for the poor extrapolation behavior of PINNs. We certainly welcome a more thorough theoretical investigation of this phenomenon, but we chose to leave that to future work.
>
> > The motivation behind the proposed transfer learning approach is not clear. I would highly appreciate it if the authors can provide the connections between the observed spectral shifts and how transfer learning can mitigate it.
>
> To summarize our motivations, we believe that by transfer learning from other PDEs that exhibit shifts in the spectra of their true solutions, the model may be able to recognize the features of PDEs that exhibit these shifting spectra and modify its predictions in the extrapolation domain accordingly. If the model has already seen the Burgers’ equation with a different viscosity parameter, for example, then this may enforce stronger inductive biases within the model that allow it to understand the evolution of the spectra for the Burger’s equation of interest.
>
> For example, our transfer learning experiments used Burgers' equations with similar viscosities ($\nu$) to the target PDE – and thus similar spectral shift, as shown in Figure 4. Transfer learning on additional PDEs, with viscosities that are further from that of the target PDE, seems to make a minimal impact. This suggests that the model may be able to map the target PDE into its shared feature space, transferring its knowledge of the PDEs that share the most similar spectral evolution to make its predictions for the target PDE.
>
> Our motivations for the proposed transfer learning approach can be found in full detail in section 4.1 ("Why does transfer learning help?").
>
> > The empirical results for improved extrapolation performance are not convincing. The L2 Extrapolation Error on the Schrodinger Equation (imag) for the proposed method is still quite poor (290%), which is still quite unusable from a practical stand-point.
>
> We agree with the reviewer that the relative L2 error in extrapolation is still too high for practical usage, even after transfer learning. The goal of this paper was not necessarily to produce results fit for practical usage but rather to put forth the idea that transfer learning may help mitigate the poor extrapolation performance induced by spectral shifts. All our analysis is conducted with standard PINNs, which naturally produce worse results than current state-of-the-art methods. We hope that applying our insights to more complex architectures/models should yield similar benefits, while proving to be more practically useful.

---

> ### Author Response · Authors · 2023-11-16
>
> > I understand the space constraints but figures 7a and b are quite important to demonstrate the results shown in Section 3.1.
>
> We agree with the reviewer that Figures 7a and 7b are helpful to contextualize the results presented in section 3.1. We have added these figures to the main text in our revision.
>
> > The placement of Figure 1 can be improved. It is referenced in Page 6 while the Figure is present in page 1. Changing the location of the figure to a more appropriate location can improve the readability of the paper.
>
> We had placed Figure 1 on page 1 to highlight our main findings, but we agree that it may be better placed later in the paper. We have modified the placement of Figure 1 (now Figure 4) accordingly in our revision, as suggested by the reviewer.
>
> > In my opinion, transfer Learning on the full domain is an unfair comparison, as the PDEs with a similar set of coefficients were already trained on the entire domain.
>
> Note that due to this shared concern, we do include results for transfer learning on the same temporal domain as the baseline for comparison; these are the Transfer (half) results in Tables 1 and 2 respectively. In both cases, we observe significant decreases in the relative L2 error of the predictions compared to baseline, though these decreases are indeed smaller than those we see when we train on the full domain.
>
> There are also instances in which our results for transfer learning from the full domain are applicable. For example, one might want to train a model on several PDEs with a vast array of different spatiotemporal domains, some of which may overlap with future target PDEs. This would of course come at some computational cost, but as seen in our transfer learning results, it could lead to improved extrapolation performance.
>
> > The authors use a different citation format than the standard ICLR format. Using the original citation format would result in overflow of the text outside the page limit since they are considerably longer.
>
> We apologize for the oversight and thank the reviewer for pointing this out – we have since modified our paper to match the standard citation format.

---

> > ### Comment · Reviewer_uE7H · 2023-11-20
> > **Response to Author Rebuttal**
> >
> > I thank the authors for providing clarifications to my questions/comments. I have read the other reviews and the authors' comments, and I have increased my score accordingly.

---

> > > ### Author Response · Authors · 2023-11-22
> > >
> > > We’d like to thank the reviewer again for their comments and feedback, and we appreciate their response to our rebuttal.

---

### Author Response · Authors · 2023-11-16
**General response**

We thank all reviewers for their careful read of our submission and the detailed feedback.

- In the revised manuscript, we have added several references to recent papers on PINNs’ generalization and extrapolation behavior to further highlight the significance of the problem to the PINN community. These changes are highlighted in red.
- Based on the reviewers’ feedback, we have also moved some figures that were previously in the appendix to the main text to make it more self-contained.
- We believe that the revisions following the reviewers’ feedback have further improved the paper. Detailed responses to the reviewers’ individual comments can be found below.

---

### Meta-Review · Area_Chair_uHko · 2023-12-13

**Metareview:**

This papers investigates the extrapolation behavior of physics-informed neural networks (PINNs). First, the authors show that the behavior depends on the underlying PDE and less on the specific design of the network architecture. Then, the authors study extrapolation in the presence of high frequency features, and conclude that high frequencies account only for a small amount of the extrapolation error. Next, they show that the extrapolation behavior mainly depends on the spectral shifts in the underlying PDE. Finally, they demonstrate that transfer learning can help to mitigate the effects of spectral shifts.

Although the paper provides several interesting observations, the findings are somewhat expected. The reviewers concur that exploring PINNs' extrapolation behavior is worthwhile but note that the empirical evidence is derived from a somewhat narrow range of PDE problems. A major weakness of this paper is the lack of theoretical insights, such as mathematical bounds, which would significantly improve the paper's value. The transfer learning study is interesting, yet it requires further exploration. Additionally, the organization of the paper can be streamlined.

While the authors have engaged in the discussion phase, not all issues could be resolved. Two reviewers remain critical, with confidence. In its present state, the paper shows promise but isn't ready for publication. It would greatly benefit from an additional iteration involving more comprehensive experiments, inclusion of theoretical insights, and improved organization. Therefore, I recommend rejecting this submission.

**Justification For Why Not Higher Score:**

In its present state, the paper shows promise but isn't ready for publication. It would greatly benefit from an additional iteration involving more comprehensive experiments, inclusion of theoretical insights, and improved organization.

**Justification For Why Not Lower Score:**

N/A

---

### Decision · Program_Chairs · 2024-01-16

Reject